# ERG-28 controls BK channel trafficking in the ER to regulate synaptic function and alcohol response in *C. elegans*

Kelly H Oh[1], James J Haney[1,2], Xiaohong Wang[3], Chiou-Fen Chuang[3,4], Janet E Richmond[4], Hongkyun Kim[1]*

[1]Department of Cell Biology and Anatomy, Chicago Medical School, Rosalind Franklin University of Medicine and Science, North Chicago, United States; [2]Department of Biology, Lake Forest College, Lake Forest, United States; [3]Division of Developmental Biology, Cincinnati Children's Hospital Research Foundation, Cincinnati, United States; [4]Department of Biological Sciences, University of Illinois at Chicago, Chicago, United States

*For correspondence: hongkyun.
kim@rosalindfranklin.edu

**Competing interests:** The authors declare that no competing interests exist.

**Abstract** Voltage- and calcium-dependent BK channels regulate calcium-dependent cellular events such as neurotransmitter release by limiting calcium influx. Their plasma membrane abundance is an important factor in determining BK current and thus regulation of calcium-dependent events. In *C. elegans*, we show that ERG-28, an endoplasmic reticulum (ER) membrane protein, promotes the trafficking of SLO-1 BK channels from the ER to the plasma membrane by shielding them from premature degradation. In the absence of ERG-28, SLO-1 channels undergo aspartic protease DDI-1-dependent degradation, resulting in markedly reduced expression at presynaptic terminals. Loss of *erg-28* suppressed phenotypic defects of *slo-1* gain-of-function mutants in locomotion, neurotransmitter release, and calcium-mediated asymmetric differentiation of the AWC olfactory neuron pair, and conferred significant ethanol-resistant locomotory behavior, resembling *slo-1* loss-of-function mutants, albeit to a lesser extent. Our study thus indicates that the control of BK channel trafficking is a critical regulatory mechanism for synaptic transmission and neural function.

## Introduction

BK channels, named for their large $K^+$ conductances, control a wide array of calcium-regulated cellular processes, including presynaptic neurotransmitter release (*Hu et al., 2001*), action potential firing rate (*Gu et al., 2007*; *Sausbier et al., 2004*; *Shao et al., 1999*), and muscle excitation (*Jaggar et al., 2000*) by limiting calcium influx. These roles of BK channels result from the unique feature that full activation of the BK channel requires coincidental membrane depolarization and elevation in free cytosolic $Ca^{2+}$ ions (*Fakler and Adelman, 2008*; *Horrigan and Aldrich, 1999*). BK channel activity is regulated by many factors including post-translational modifications, associating auxiliary subunits, its localization at the plasma membrane, and the number of channels expressed at the plasma membrane (*Kim and Oh, 2016*). Despite its importance, how BK channel density at the plasma membrane is determined is not well understood. Several proteins have been reported to affect the surface membrane expression of BK channels. For instance, cytoskeletal proteins and cytoskeleton-interacting proteins, including actin, microtubule-associated protein 1A, and filamin A, are reported to alter the level of BK channels at the plasma membrane in vitro (*Kim et al., 2007*; *Park et al., 2004*; *Tian et al., 2006*). A recent study showed that cereblon mediates the interaction of BK channels with CRL4A E3 ubiquitin ligase complex, which causes ubiquitination of BK channels

and their retention in the ER (*Liu et al., 2014*). Blocking BK channel ubiquitination or disrupting the interaction between BK channels and CRL4A E3 ubiquitin ligase was shown to increase the surface expression of BK channels in vitro in transfected primary cultured neurons. However, most of the studies on BK channel trafficking have been performed in heterologous cells that are designed to overexpress BK channels. For this reason, whether proteins that are reported to influence BK channel trafficking in vitro alter BK channel levels in vivo has not been clearly addressed.

In *C. elegans*, BKIP-1, a plasma membrane localized protein, was shown to increase the levels of SLO-1 BK channels at the plasma membrane (*Chen et al., 2010*). In addition to the expression level at the plasma membrane, SLO-1 localization at specific membrane microdomains also plays an important role in determining SLO-1 function. *C. elegans* genetic studies identified genes that control the localization of SLO-1 channels in both muscles and neurons: The dystrophin complex localizes SLO-1 channels near calcium channels in the sarcolemma of body wall muscle (*Kim et al., 2009*). It was also shown that alpha-catulin and dystrobrevin hierarchically organize BK channels near calcium channels at presynaptic terminals (*Abraham et al., 2010*; *Oh et al., 2015*).

Biogenesis of plasma membrane proteins begins in the endoplasmic reticulum (ER). Once completely folded and assembled, most plasma membrane proteins are packaged in cargo vesicles that exit from the ER through the COPII-dependent mechanism (*Zanetti et al., 2012*). It appears that proteins with multiple transmembrane domains require ER membrane chaperons and cargo adaptor proteins for their efficient exit from the ER. For example, yeast Erv14 is a cargo receptor that links many client membrane proteins to the COPII coat adaptor and is required for the efficient trafficking of many polytopic membrane proteins (*Herzig et al., 2012*). It is not known if the BK channel requires an ER chaperone, or a cargo receptor for its efficient trafficking. In our present study, we identified an ER membrane protein that promotes BK channel expression at the plasma membrane.

In a *C. elegans* genetic screen designed to identify genes that regulate SLO-1 function, we identified the *erg-28* gene that is homologous to yeast and mammalian ERG28. We found that ERG-28 localizes to the endoplasmic reticulum membrane and is required for the trafficking of SLO-1 to the plasma membrane. In the absence of ERG-28, SLO-1 is not efficiently trafficked to the plasma membrane and undergoes degradation. Accordingly, *erg-28* mutants exhibit phenotypes of *slo-1* loss-of-function mutants including ethanol-resistant locomotory behavior. Thus, ERG-28 offers a new regulatory point that controls the synaptic abundance of SLO-1 and thus neurotransmitter release.

## Results

### A mutation in *erg-28*, a gene distantly related to yeast ergosterol biosynthesis 28, suppresses phenotypic defects of *slo-1* gain-of-function mutant

In a genetic screen designed to identify mutants that suppress the sluggish movement of a *slo-1* gain-of-function mutant, *slo-1(ky399gf)*, we identified *cim16* mutant. Compared to *slo-1(ky399gf)*, *cim16; slo-1(ky399gf)* double mutants showed improved movement (*Figure 1A*). Importantly, the speed of *cim16* mutants was comparable to that of wild-type animals, indicating that the improved locomotory speed of the *cim16;slo-1(ky399gf)* double mutant is not the result of an overly active locomotory phenotype of *cim16*.

By a combination of SNP (single nucleotide polymorphism)-based genetic mapping and transformation rescue (*Figure 1—figure supplement 1*), we identified the *cim16* mutation in *erg-28* gene. *erg-28* is an orthologue of the yeast ergosterol biosynthesis 28 (Erg28) (*Vinci et al., 2008*). The ERG28 protein sequence has diverged in higher organisms (*Veitia and Hurst, 2001*). These homologues possess four predicted transmembrane domains instead of the two transmembrane domains found in yeast and have acquired the ER retention/retrieval motif in the C-terminus (*Figure 1B*). Although *C. elegans* ERG-28 retains four transmembrane domains and the ER retention/retrieval motif, it is further diverged from mammalian ERG28 proteins; it shows 22.2% identity (56.4% similarity) with the human homolog C14orf1 based on a non-overlapping local alignment (*Figure 1—figure supplement 1*). A nonsense mutation in *cim16* leaves the coding sequence intact, except for seven amino acids at the C-terminus (*Figure 1B*). To ascertain that the genetic defect in *erg-28* is responsible for the suppression of the *slo-1(ky399gf)* phenotype rather than a background mutation, we

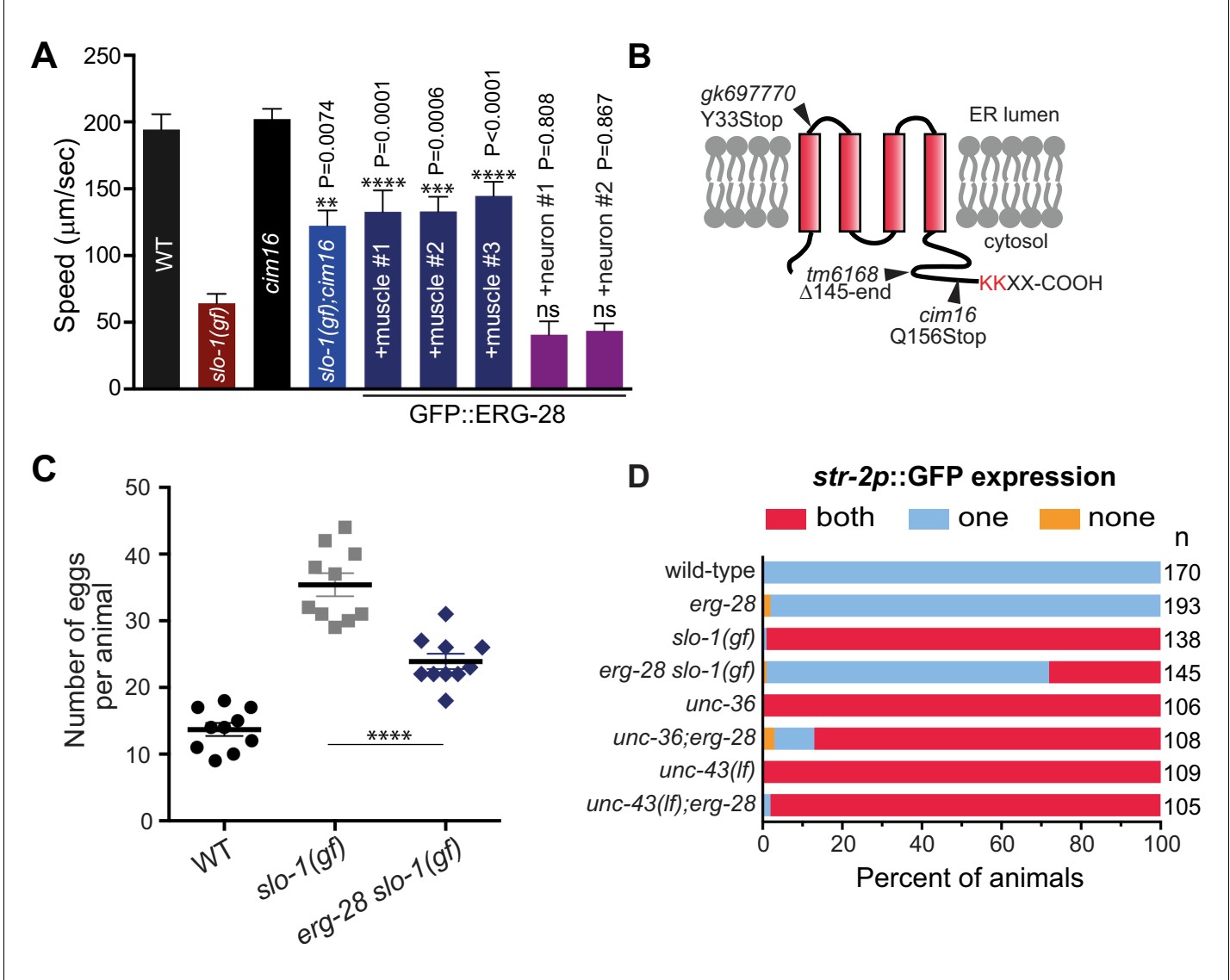

**Figure 1.** Loss-of-function mutation in *erg-28* suppresses phenotypic defects of *slo-1* gain-of-function mutant. (A) Mutation in *erg-28* suppresses sluggish locomotion of *slo-1(ky399gf)* mutant. Average speed of animals with indicated genotypes on NGM agar plates without food. *slo-1(gf)* and *erg-28* alleles are *ky399* and *cim16*, respectively. +muscle #1, #2, #3: three independent lines of *erg-28 slo-1(gf);cimEx[myo-3p::gfp::erg-28, ttx-3p::mRFP]*. +neuron#1, #2: two independent transgenic lines of *erg-28 slo-1(gf);cimEx[H20p::gfp::erg-28, ttx-3p::mRFP]*. Error bars represent S.E.M. (standard error of the mean). n = 10. One way ANOVA with Tukey's post-hoc analysis. P values are presented in comparison to *slo-1(gf)*. (B) ERG-28 is predicted to be an endoplasmic reticulum resident protein with four transmembrane domains and KKXX ER retrieval COPI binding motif. Mutation sites of *cim16*, *tm6168*, and *gk697770* alleles are indicated. (C) Suppression of egg laying defects of *slo-1(ky399gf)* mutant by *erg-28(cim16)* mutation. Number of eggs retained in the animals. Error bars represent S.E.M. n = 10. ****p<0.0001, one way ANOVA with Tukey's post-hoc analysis. (D) Suppression of defective asymmetric *str-2* gene expression in *slo-1(ky399gf)* mutant by *erg-28(gk697770)*. Mutants used in this experiment: *erg-28(gk697770)*, *slo-1(ky399)*, *unc-36 (e251)*, and *unc-43(n498n1186)*. (*Figure 1—source data 1*: This file contains raw measurement data and statistical analysis summary for *Figure 1A, C and D*).

The following source data and figure supplements are available for figure 1:

**Source data 1.** *erg-28* mutation suppresses phenotipic defects of *slo-1(gf)* mutant.

**Figure supplement 1.** Identification of *cim16* mutation in *erg-28* gene.

**Figure supplement 2.** Both *gk697770* and *tm6168* alleles of *erg-28* suppress the sluggish movement of *slo-1(ky399)* gain-of-function mutant.

*Figure 1 continued on next page*

*Figure 1 continued*

**Figure supplement 2—source data 1.** Two other alleles of *erg-28* suppress the locomotory defect of *slo-1(gf)* mutant.

**Figure supplement 3.** The human homolog C14orf1 (herg-28) can partially replace *C. elegans erg-28*.

**Figure supplement 3—source data 1** Functional conservation of human homolog with *C. elegans erg-28*.

tested two other independently isolated alleles, *erg-28(gk697770)* and *erg-28(tm6168)*, which were generated by the Million Mutation Project (*Thompson et al., 2013*) and the National BioResource Project, respectively. The *erg-28(gk697770)* allele has a nonsense mutation in the 33rd amino acid residue and the *erg-28(tm6168)* allele is a deletion/insertion allele that introduces a nonsense codon at 145th amino acid residue in addition to the deletion (*Figure 1B*). Both alleles also suppressed the sluggish movement of *slo-1(ky399gf)* mutants to a level similar to *cim16* (*Figure 1—figure supplement 2*). Although the phenotypes of the three alleles are indistinguishable, the *erg-28(gk697770)* allele is predicted to be most severe based on the location of its nonsense mutation site. Thus, most of the subsequent experiments were performed using *erg-28(gk697770)* allele.

Our previous studies showed that neuronal function of *slo-1* is responsible for locomotory speed (*Abraham et al., 2010*). We tested whether neuronal expression of *erg-28* restores the sluggish movement of *slo-1(ky399gf)* by expressing GFP-tagged ERG-28 in *erg-28(cim16) slo-1(ky399gf)* double mutants under the control of the pan-neuronal *H20* promoter (*Shioi et al., 2001*) or muscle specific *myo-3* promoter. Neuronal, but not muscle, expression of *erg-28*, reverted the speed of *erg-28 (cim16) slo-1(ky399gf)* double mutants to sluggish movement (*Figure 1A*), indicating that *erg-28* functions in neurons to regulate SLO-1 function in locomotion. Given this neuronal function of *erg-28*, we also determined whether the human homolog C14orf1 can replace *C. elegans erg-28* function by expressing C14orf1 under the control of the pan-neuronal *H20* promoter in *erg-28(cim16) slo-1 (ky399gf)* double mutants. We found that three out of four independent transgenic lines partially restored the sluggish movement of *slo-1(ky399gf)*, suggesting its functional conservation with *C. elegans* ERG-28 (*Figure 1—figure supplement 3*).

In addition to the sluggish locomotion, *slo-1(ky399gf)* mutants also exhibit egg laying defects (*Oh et al., 2012*) and loss of asymmetric gene expression in AWC olfactory neuron pairs (*Troemel et al., 1999*). We examined if *erg-28* mutation suppresses these phenotypes as well. *erg-28* mutation was partially able to suppress the egg laying defect, as *erg-28(gk697770) slo-1(ky399gf)* double mutants retained less eggs than the *slo-1(ky399gf)* single mutant (*Figure 1C*). A pair of AWC olfactory neurons differentiates into two distinctive subtypes; one that expresses the *str-2* gene (AWC$^{ON}$) and the other that does not (AWC$^{OFF}$). Contrary to wild-type animals, over 90% of the *slo-1(ky399gf)* mutant animals express the *str-2* gene in both of the AWC neurons (2 AWC$^{ON}$). *erg-28* mutation was able to suppress this *slo-1(ky399gf)* phenotype, as less than 30% of the *erg-28 (gk697770) slo-1(ky399gf)* double mutant animals express *str-2* genes in both AWC neurons (*Figure 1D*). The asymmetric *str-2* gene expression is governed by a calcium signaling pathway involving *unc-2/unc-36* voltage-gated calcium channels and downstream CaMKII, *unc-43* (*Alqadah et al., 2016*; *Schumacher et al., 2012*). *erg-28* mutation was not able to suppress the 2 AWC$^{ON}$ phenotype of the *unc-43* mutant and minimally suppressed the *unc-36* mutant (*Figure 1D*). These results place *erg-28* upstream of *unc-43* and parallel to *unc-2/unc-36,* suggesting *erg-28* is most likely to act on *slo-1*. Taken together we conclude that *erg-28* mutation suppresses the phenotypic defects of the *slo-1(ky399gf)* mutant in general and mutation in *erg-28* is most likely to result in a reduction of SLO-1 function.

## ERG-28 is required for SLO-1 function in synaptic transmission

Previous studies showed that SLO-1 channels provide a negative feedback response for calcium influx and thus negatively regulate synaptic transmission. Our previous study showed that *slo-1* loss-of-function mutants exhibit a prolonged evoked synaptic response, whereas *slo-1(ky399gf)* mutants

exhibit a reduced evoked synaptic response (*Abraham et al., 2010*; *Sancar et al., 2011*). Hence, we reasoned that if the underlying mechanism for the suppression of *slo-1 (ky399gf)* by *erg-28* mutation is reduction of SLO-1 function, it is expected that *erg-28* mutant would exhibit increased synaptic transmission. First, we sought to determine the pharmacological sensitivity to aldicarb, an acetylcholinesterase inhibitor. While mutant animals with decreased levels of cholinergic synaptic transmission are resistant to aldicarb-mediated paralysis, mutants with increased levels of synaptic transmission are hypersensitive to aldicarb (*Mahoney et al., 2006*). We found that *erg-28(gk697770)* animals are hypersensitive to aldicarb. Furthermore, *erg-28* mutation suppresses the aldicarb-resistant phenotype of *slo-1(ky399gf)* mutants (*Figure 2A*, *Figure 2—source data 1*). Next, we employed electrophysiology to further examine the effect of *erg-28* mutation on SLO-1 function in synaptic transmission. We compared evoked current responses at the neuromuscular junctions of wild-type, *erg-28(gk697770)*, *slo-1(ky399gf)*, and *erg-28(gk697770) slo-1(ky399gf)* double mutant animals. We did not observe a higher evoked response in *erg-28* relative to wild-type animals, whereas *slo-1 (ky399gf)* animals showed a reduced evoked response and faster decay kinetics compared to wild-type and *erg-28* animals (*Figure 2B and C*). *erg-28* mutation significantly improved the reduced evoked response of *slo-1(ky399gf)* mutants, but not decay kinetics (*Figure 2B–E*). Together, these results support the hypothesis that ERG-28 reduces SLO-1 synaptic function.

## ERG-28 is an ER resident membrane protein expressed in neurons and muscles

The *erg-28* gene forms an operon with *C14C10.5*, which encodes a homolog of proteasome activator subunit 4 (PSME4) (*Figure 1—figure supplement 1*). The shared promoter region, 2.5 kb upstream of *C14C10.5*, was fused to GFP coding sequence, and the resulting construct was used to determine the tissue expression pattern of *erg-28*. The expression of *erg-28* was found in many different tissues, including muscles, intestine, and neurons (*Figure 3A*). Promoters of operons are often promiscuous and expression of individual genes in the operons is controlled at a post-transcriptional level (*Blumenthal, 2005*; *Blumenthal et al., 2015*). Thus, to evaluate *erg-28* gene expression, we generated an integrated transgenic line that expresses mCherry-tagged ERG-28 driven by the shared 2.5 kb promoter. In this transgenic line, ERG-28 expression is observed mainly in neurons and muscles, where SLO-1 is predominantly expressed (*Figure 3B*). After establishing its expression pattern, we analyzed the subcellular localization of ERG-28 in neurons by expressing GFP-tagged ERG-28 in a subset of motor neurons (9 DA and DB cholinergic motor neurons) using 2.4 kb *unc-129* promoter along with the ER and Golgi markers. Localization of GFP-tagged ERG-28 most likely reflects endogenous ERG-28 localization, since GFP-tagged ERG-28 was able to rescue the loss of ERG-28 (*Figure 1A*). Cell bodies and dendrites of the DA and DB neurons are located in the ventral cord, and their axons are located in the dorsal cord, thus allowing us to distinguish axons from dendrites (*Colavita et al., 1998*). We found that ERG-28 is abundantly expressed proximal to neuronal nuclei and localized as distinct puncta throughout the synaptic terminals and dendrites (*Figure 3C*). ERG-28 puncta co-localize well with the ER marker (mCherry::PISY-1) and exhibit minimal co-localization with the Golgi marker (AMAN-2::mCherry) (*Figure 3D*). These results are consistent with previous studies that the ER extends to axonal and dendritic neural processes in *C. elegans* (*Brockie et al., 2013*; *Rolls et al., 2002*). Importantly, the distribution of ERG-28 is not restricted to axon or dendrites, indicating that ERG-28 does not have axon-dendrite polarity. Like mammalian ERG28, the *C. elegans* ERG-28 C-terminus has a well-established consensus ER retrieval/retention motif (KKXX-COOH), which interacts with coat protein complex I (COPI) that mediates retrograde transport of target proteins from the *cis*-end of the Golgi complex to the ER, thereby ensuring the ER localization of the target proteins (*Jackson, 2014*). When GFP-tagged *erg-28Δ7* lacking 7 amino acids at the C terminus as in *cim16* allele was expressed in the DA and DB motor neurons, the overall expression was significantly decreased and normal punctate structures were no longer visible (*Figure 3E*). We cannot determine if the functional defect of the *cim16* allele is due to a structural instability independently of defective ER localization or localization defect caused by the absence of the ER retrieval sequence. Since the *cim16* allele suppresses *slo-1(ky399gf)* allele as effectively as *gk697770* allele (*Figure 1—figure supplement 2*), it is very likely that the C-terminal 7 amino acids are important for the stable expression and localization of ERG-28. The presence of the COPI binding motif in ERG-28 strongly suggests that ERG-28 shuttles between ER and Golgi. Indeed, we find

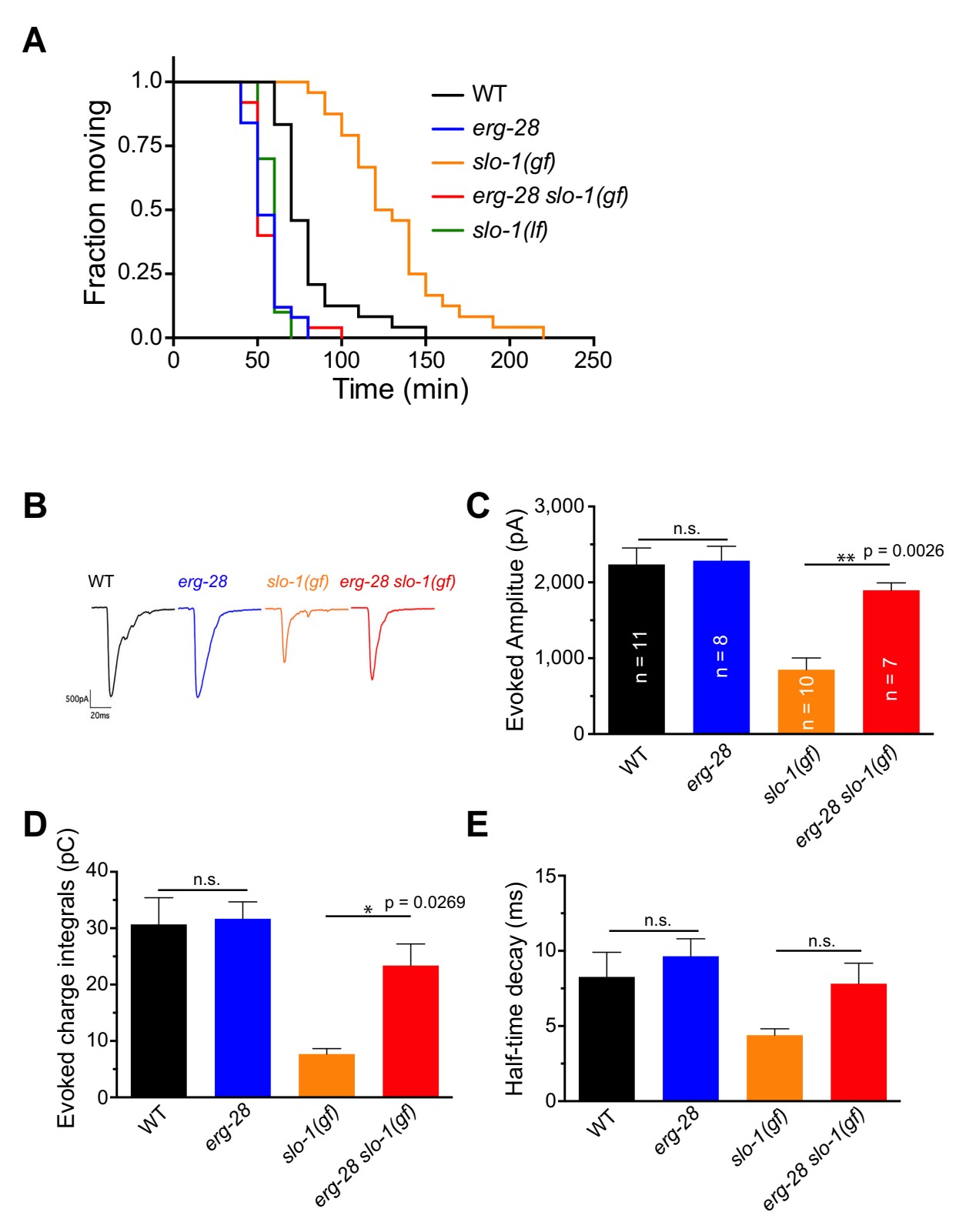

**Figure 2.** ERG-28 is required for SLO-1 function in synaptic transmission. (A) *erg-28(gk697770)* mutation suppresses aldicarb resistance of *slo-1(ky399gf)* mutant and is aldicarb hypersensitive. Aldicarb-induced paralysis was analyzed using Kaplan-Meier survival analysis. Three independent trials with n > 20 for each genotype in each trial. All three trials showed similar results. Trial 1 is shown and statistics of 3 trials is presented in *Figure 2-source data 1*. Mutants used are *erg-28 (gk697770)*, *slo-1(ky399)*, and *slo-1(eg142 loss of function)*. 1 mM aldicarb was used. (B) Representative evoked

*Figure 2 continued on next page*

*Figure 2 continued*
responses from voltage-clamped body wall muscles (holding potential −60 mV) in response to 2 ms blue-light activation of neuronally expressed channelrhodopsin-2. (**C–E**) *erg-28(gk697770)* suppresses the synaptic defects of *slo-1(ky399gf)* mutants. n.s.: not significant, One way ANOVA with Tukey's post-hoc analysis. *Figure 2—source data 1*: This file contains raw measurement data and statistical analysis summary for *Figure 2C,D and E*.
The following source data is available for figure 2:

**Source data 1.** *erg-28* mutation suppresses the synaptic transmission defect of *slo-1(gf)* mutant.

that some of the ERG-28 puncta are mobile (*Video 1*). These observations suggest that ERG-28 is localized to the ER membrane through COPI interaction.

To further confirm that ERG-28 is localized in ER at steady state, we examined its localization in body wall muscle cells, which are much larger than neurons (*Figure 3F and G*). We expressed GFP-tagged ERG-28 along with either the ER marker mCherry-tagged PISY-1 or the Golgi marker mCherry-tagged AMAN-2 in muscle cells. Because GFP-tagged ERG-28 is functional (*Figure 1A*), it is likely that GFP-tagged ERG-28 reflects in vivo localization of ERG-28. The ER aligns around the dense body and is closely apposed to the plasma membrane in the myofilament lattices, and exhibits a mesh-like network in the non-contractile muscle belly, where cellular organelles, including the nucleus and mitochondria, are present (*Altun and Hall, 2009*). GFP-tagged ERG-28 well co-localizes with mCherry-tagged PISY-1 from the muscle interior (muscle belly) to the muscle plasma membrane (*Figure 3F*, *Figure 3—figure supplement 1*). By contrast, mCherry-tagged AMAN-2 shows a completely different pattern from that of mCherry-tagged PISY-1 and does not co-localize with GFP-tagged ERG-28 (*Figure 3G*, *Figure 3—figure supplement 1*). These results together indicate that ERG-28 localizes to the ER.

## ERG-28 is required for the normal expression level of SLO-1 at presynaptic terminals and muscle excitation sites

To understand the mechanism underlying the reduced *slo-1* function observed in *erg-28* mutants, we first considered the ergosterol synthesis pathway. Yeast ERG28 is a gene necessary for ergosterol (a cholesterol derivative) biosynthesis (*Mo and Bard, 2005*; *Mo et al., 2004*) and functions as a molecular scaffold for sterol biosynthetic enzymes in ER membranes. Nematodes (*C. elegans*) and arthropods (*Drosophil*a) cannot synthesize sterol de novo because of the lack of key biosynthetic enzymes. They must therefore obtain sterol via dietary intake. Nonetheless, *C. elegans* possesses a few genes that are homologous to yeast ERG28p-binding ergosterol biosynthetic enzyme genes. We found that unlike *erg-28*, none of the tested mutations in these candidate biosynthetic genes (*erg-24 (tm3499)*, *erg-26 (tm1941)*, *erg-6 (tm1781)*) suppressed the sluggish movement of *slo-1* (*ky399gf*) (**Data not shown**), raising the possibility that the suppression of the *slo-1(gf)* phenotype by *erg-28* does not occur through the modulation of sterol synthesis pathway per se, although it is possible that these genes function redundantly in modulating cholesterol.

Next, we considered if *erg-28* mutation affects SLO-1 expression and localization in neurons and muscles similar to other genetic suppressors of *slo-1(ky399gf)* that we and others had previously isolated. To measure the SLO-1 expression and localization in vivo, we used the CRISPR/Cas9 genome-editing technology to fuse GFP to the C-terminus of SLO-1 (*Figure 4—figure supplement 1*). The edited line, *cim105[slo-1::GFP]*, designed to tag all of 12 SLO-1 isoforms (*Johnson et al., 2011*), does not show any obvious behavioral phenotypes found in *slo-1* loss-of-function mutants, such as exaggerated head bending during forward movement, and its locomotory speed was also comparable to that of the wild type animals (*Figure 4—figure supplement 2A–C*), indicating that GFP fusion to the C-terminus of SLO-1 in *cim105* animals does not perturb in vivo SLO-1 function. While weak, diffused SLO-1 channels can be detected in the nerve ring, SLO-1 channels are mainly observed in neurons and muscles as discrete puncta (*Figure 4—figure supplement 2D*).

When we compared SLO-1::GFP expression in wild-type, *erg-28 (gk697770)*, and *erg-28(cim16)* mutant animals by taking advantage of *cim105[slo-1::GFP]*, SLO-1::GFP expression was drastically reduced in *erg-28* mutant animals both in the nerve ring and muscles, although not completely abolished (*Figure 4A*). In wild-type muscles, SLO-1::GFP showed the same puncta pattern as previously reported using transgenic animals (*Abraham et al., 2010*). In *erg-28(gk697770)* and *erg-28(cim16)*

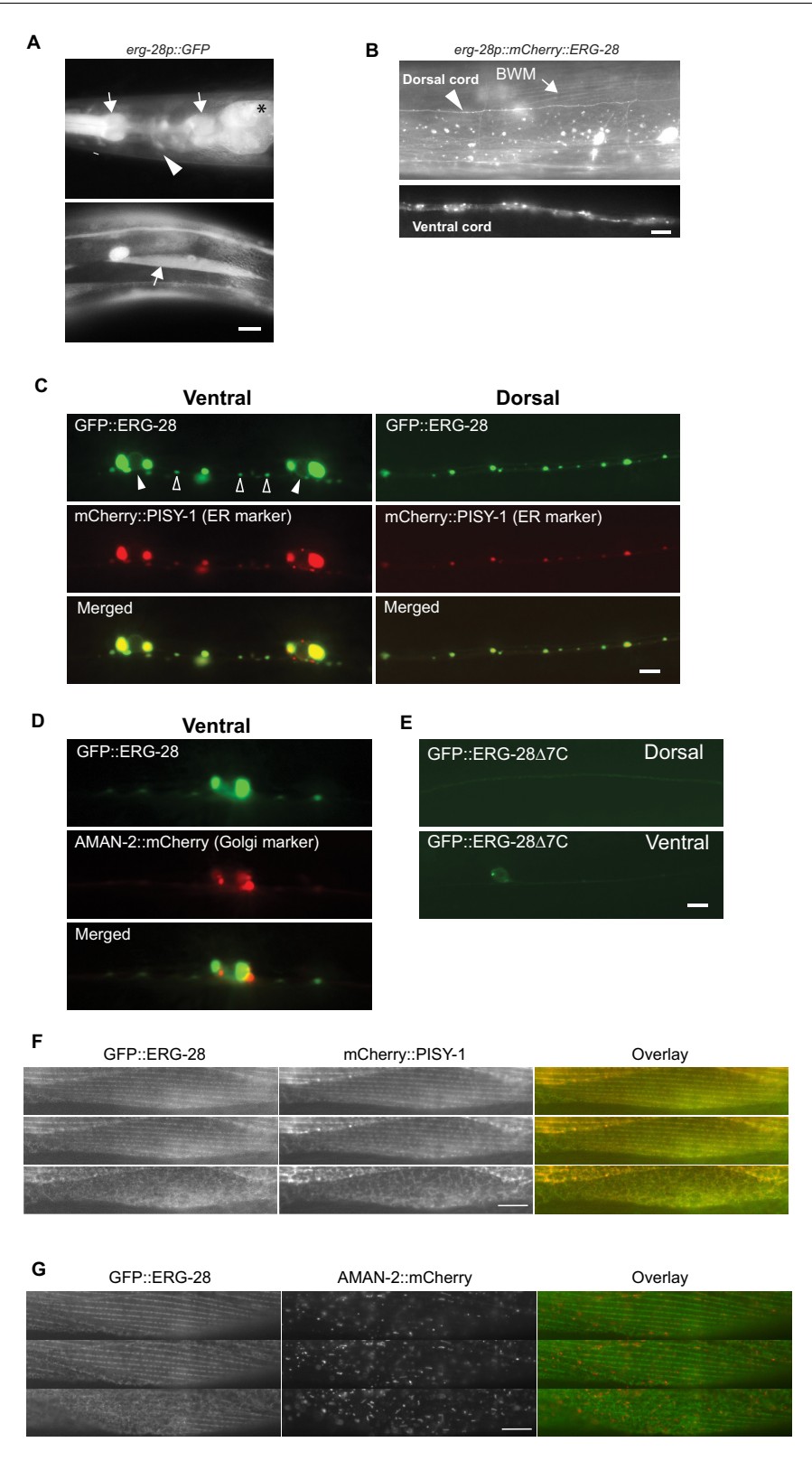

**Figure 3.** ERG-28 is an endoplasmic reticulum resident protein expressed in muscles and neurons. (**A**) *erg-28* is a part of an operon with *C14C10.5*. 2.5 kb upstream of *C14C10.5* gene including 30 bp of coding region of *C14C10.5* was fused with GFP (*cimEx54[erg-28p::gfp, ttx-3p::mRFP]*). Expression was observed in muscles (arrows), neurons (arrowheads), and intestines (asterisk). Scale bar: 10 μm. (**B**) ERG-28 is prominently expressed in muscles (arrow) and neurons (arrowhead). mCherry-tagged ERG-28 (*cimIs39[erg-28p::mCherry::erg-28, odr-1p::gfp]*) was expressed under the same promoter as

*Figure 3 continued on next page*

*Figure 3 continued*

in **Figure 3A**. Scale bar: 10 µm. (**C**) ERG-28 co-localizes with the ER marker PISY-1 in a subset of motor neurons. GFP::ERG-28 and PISY-1::mCherry were expressed using the *unc-129* neuronal promoter (*cimEx63[unc-129p::mCherry::pisy-1, unc-129p::gfp::erg-28]*). Closed arrowheads: neuron nuclei. Open arrowheads: puncta at dendrites. Scale bar: 10 µm. (**D**) ERG-28 is minimally co-localized with the Golgi marker AMAN-2 in a subset of motor neurons. GFP::ERG-28 and AMAN-2::mCherry were expressed using the *unc-129* neuronal promoter (*cimEx59[unc-129p::aman-2::mCherry, unc-129p::gfp::erg-28]*). Scale bar: 10 µm. (**E**) The last 7 amino acids of ERG-28 are required for stable expression of ERG-28. ERG-28::GFP with 7 amino acid deletion was expressed using *unc-129* neuronal promoter (*cimEx66[unc-129p::gfp::erg-28Δ7, ttx-3p::mRFP]*). Scale bar: 10 µm. (**F**) GFP::ERG-28 co-localizes with the ER marker in muscle cells. GFP::ERG-28 and mCherry::PISY-1 were expressed using the *myo-3* promoter (*cimEx88[myo-3p::gfp::erg-28, myo-3p::mCherry::pisy-1, rol-6(d)]*). Scale bar: 10 µm. Three different sections from the z-stack are shown. (**G**) GFP::ERG-28 does not co-localize with the Golgi marker in muscle cells. GFP::ERG-28 and AMAN-2::mCherry were expressed using the *myo-3* promoter (*cimEx91[myo-3p::gfp::erg-28, myo-3p::aman-2::mCherry, rol-6(d)]*). Scale bar: 10 µm. Three different sections from the z-stack are shown. Images of additional sections are shown in **Figure 3—figure supplement 1**.

The following figure supplement is available for figure 3:

**Figure supplement 1.** GFP::ERG-28 co-localizes with the ER marker (**A**), but not with the Golgi marker (**B**) in muscle cells.

mutants, a similar puncta pattern was observable, although the expression level was drastically reduced (**Figure 4A**). When measured by western analysis using whole animals, the total amount of SLO-1 was reduced in *erg-28* mutants, confirming the overall reduction of SLO-1 amount (**Figure 5D**). To further confirm that the reduction of SLO-1 is due to *erg-28* mutations, we examined if transgenic expression of ERG-28 would restore SLO-1 expression at the plasma membrane. We introduced an integrated transgenic array that expresses mCherry-tagged ERG-28 under the control of its own promoter into *erg-28 (gk697770) slo-1(cim105)* mutant animals and determined SLO-1 expression levels. The expression of mCherry-tagged ERG-28 restored the SLO-1::GFP expression to a level comparable to the heterozygotic expression of SLO-1::GFP (one copy of untagged SLO-1 and one copy of GFP-tagged SLO-1) in both neurons and muscles (**Figure 4B and C**).

Our previous study showed that transgenically expressed SLO-1 is localized at the presynaptic terminals (**Oh et al., 2015**). To examine if SLO-1::GFP puncta in *erg-28* mutant are correctly localized at the presynaptic terminals as in the wild type animals, we generated a line that expresses fRFP fused ELKS-1, an active zone marker, by genome editing (**Figure 4—figure supplement 1**). First, the genome edited line, *cim107[elks-1::fRFP]*, was crossed with *cim105* and we examined whether SLO-1::GFP co-localizes with ELKS-1::fRFP. We find that SLO-1::GFP and ELKS-1::fRFP are well co-localized although we occasionally observed that some SLO-1::GFP puncta that did not co-localize with ELKS-1::fRFP, and *vice versa* (**Figure 4D**). While we and others previously observed broad SLO-1 expression in cell bodies of neurons with integrated transgenic lines (**Chen et al., 2010**; **Oh et al., 2015**), we did not observe such expression in the cell bodies with *cim105[slo-1::GFP]* animals (**Figure 4D**). In the *erg-28* mutant, the SLO-1 puncta were co-localized with ELKS-1::fRFP (**Figure 4E**). These results indicate that ERG-28 is required for the normal level of SLO-1 expression in muscles and neurons, but not for the localization at either presynaptic terminals or dense bodies in muscles.

## ERG-28 promotes SLO-1 trafficking to the plasma membrane

At the steady state level, ERG-28 is localized in the ER and SLO-1 is at the plasma membrane. How is an ER resident protein required for the normal expression of a plasma membrane localized protein? As a plasma membrane-localized

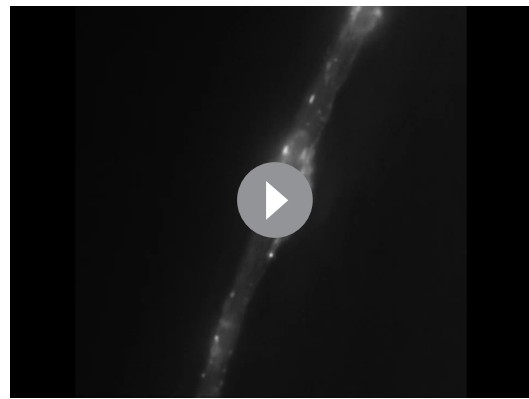

**Video 1.** ERG-28 is mobile. mCherry::ERG-28 was expressed using *erg-28* promoter, *cimIs39[erg-28p::mCherry::erg-28, odr-1p::gfp]*. Time lapse image. A movie recorded for 14 s.

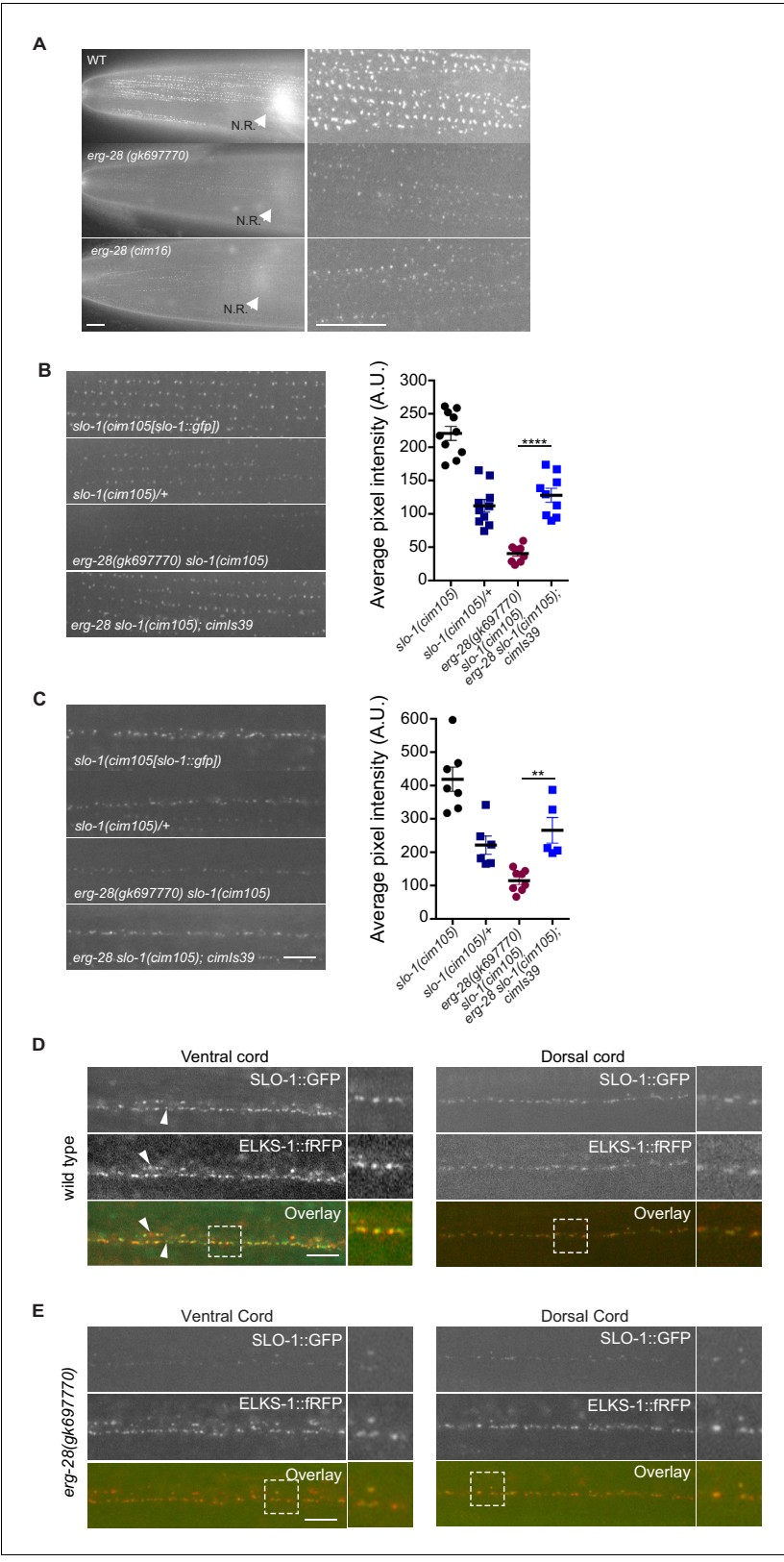

**Figure 4.** ERG-28 is required for the normal expression level of SLO-1 at synaptic terminals and muscle excitation sites. (A) SLO-1::GFP expression is reduced in *erg-28(gk697770)* and *erg-28(cim16)* mutants. N.R: nerve ring. Scale bar: 10 μm. (B and C) SLO-1 expression is restored by expression of ERG-28 in muscle cells (B) and neurons (C). Right panels are quantification of puncta in muscles (B) and the dorsal cords (C). *slo-1(cim105)/+* heterozygote animals were generated by crossing *slo-1(cim105)* with wild-type males. *cimIs39[erg-28p::mCherry::erg-28, odr-1p::gfp]* is the
*Figure 4 continued on next page*

*Figure 4 continued*

chromosomally integrated array expressing mCherry::ERG-28 from the *erg-28* promoter. (D) SLO-1::GFP is expressed at presynaptic terminals and co-localizes with active zone marker, ELKS-1::fRFP in the nerve cords. Occasionally, puncta that do not co-localize with each other are observed (arrowheads). Dash lined boxes indicate the areas that are zoomed in and shown in the right. Scale bar: 10 μm. (E) In *erg-28(gk697770)* mutant, SLO-1::GFP expression is drastically reduced, but the remaining SLO-1::GFP co-localizes with ELKS-1::fRFP in the nerve cords. Scale bar: 10 μm.

The following source data and figure supplements are available for figure 4:

**Source data 1.** SLO-1 expression in the muscle (B) and neuron (C) is reduced in *erg-28* mutant.

**Figure supplement 1.** Introduction of GFP and fRFP (FusionRed) into the *slo-1* (A) and *elks-1* (B) genomic loci using CRISPR/Cas9 technology.

**Figure supplement 2.** Introduction of GFP at the end of the genomic *slo-1* coding sequence does not interfere with head bending angle (A and B) or normal locomotory speed (C).

**Figure supplement 2—source data 1.** The C-terminal GFP fusion does not interfere with normal function of SLO-1.

protein, SLO-1 must be trafficked through the ER and Golgi complex to its final location at the plasma membrane. Thus, ERG-28 is likely to play a role during the biogenesis of SLO-1 in the ER or transport from the ER to the Golgi complex. Given their steady state localization and the mobile nature of ERG-28, the interaction between ERG-28 and SLO-1 is expected to be transient, if it occurs at all. Indeed, we find that SLO-1::GFP puncta rarely co-localize with mCherry::ERG-28 puncta although potential co-localization can be occasionally observed in neuronal cell bodies (*Figure 5—figure supplement 1*). Taken together, ERG-28 is most likely to function in ER for the efficient trafficking of SLO-1 to the plasma membrane.

If ERG-28 is required for SLO-1 transport from the ER to the Golgi complex, we would expect to see an accumulation of SLO-1 in the ER in *erg-28* mutants. However, SLO-1 accumulation in the ER was not observed in *erg-28* mutants. Thus, we hypothesized that in the absence of ERG-28, SLO-1 is degraded by the proteasome. To test this hypothesis, we examined whether treatment with a proteasome inhibitor (bortezomib) would increase SLO-1 expression in the ER or at the plasma membrane. When *erg-28* mutant animals were grown in the presence of bortezomib, SLO-1 expression at presynaptic terminals, but not in cell bodies, was increased by 50%, while the wild type animals showed no statistically significant change (*Figure 5A*). This suggests that ERG-28 protects SLO-1 from degradation in the ER and is not an obligatory factor for SLO-1 exit from the ER. Thus, the lower expression level of SLO-1 at the plasma membrane in *erg-28* mutant is at least in part due to proteasomal degradation.

To identify genes responsible for the SLO-1 degradation in *erg-28* mutants, we tested some of the genes implicated in the ERAD (ER associated degradation) and proteasomal degradation pathway to determine whether their reduction of function would increase SLO-1 expression in *erg-28 (gk697770)* mutant. From this approach, we identified *ddi-1* as a gene that participates in the degradation of SLO-1 in *erg-28* mutants. *ddi-1* is an orthologue of yeast and human DDI1, which is thought to be a shuttling factor that delivers ubiquitinated substrates to proteasomes (*Gabriely et al., 2008*). In yeast, DDI-1 appears to act as a negative regulator for general exocytosis (*White et al., 2011*). Compared to wild type, the *ddi-1(ok1468)* single mutant did not show any significant difference in SLO-1 expression level in neurons and muscles. However, *ddi-1(ok1468);erg-28 (gk697770)* double mutants showed a significantly higher expression of SLO-1 at the plasma membrane than the *erg-28(gk697770)* single mutant, albeit to a lesser extent than wild-type animals (*Figure 5B–D*). Taken together, we conclude that ERG-28 promotes SLO-1 channel trafficking to the plasma membrane by shielding SLO-1 channels from proteasomal degradation.

## ERG-28 is required for normal alcohol response in *C. elegans*

Previous studies have shown that SLO-1 BK channels are a central mediator of alcohol intoxication (*Davies et al., 2003*; *Davis et al., 2014*; *Dopico et al., 1996*). Given that *erg-28* mutation reduces

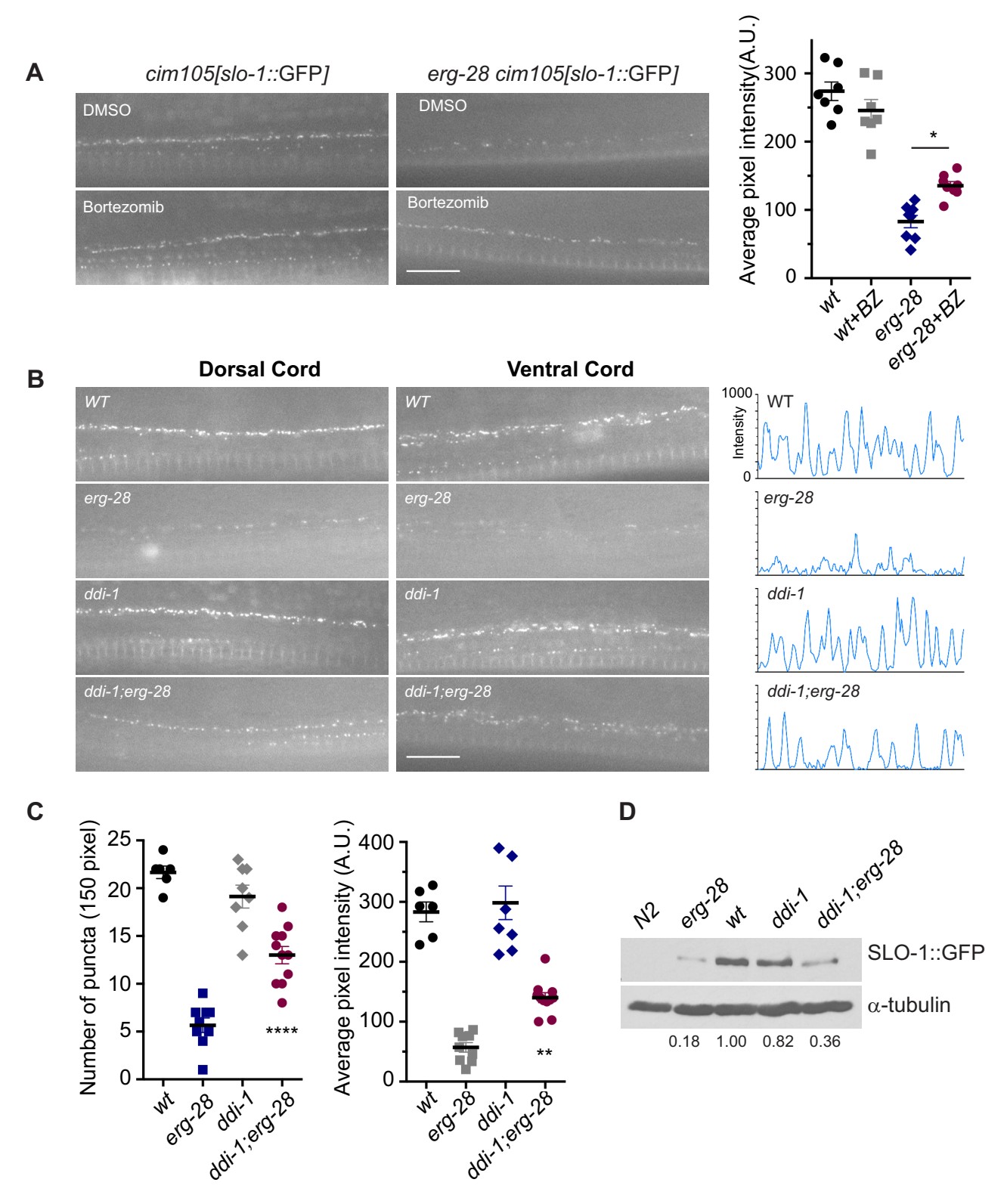

**Figure 5.** SLO-1 is degraded by proteasome in the absence of ERG-28. (**A**) Proteasome inhibition with bortezomib (BZ) partially restores SLO-1 expression at the presynaptic terminals in *erg-28(gk69777)* mutant. Scale bar: 10 μm. Right panel: SLO-1 average pixel intensity in the dorsal cord of DMSO or BZ treated strains was quantified. Error bars represent S.E.M. *p=0.0112, one way ANOVA with Tukey's post-hoc analysis. (**B**) *ddi-1* mutation partially restores SLO-1 expression in *erg-28* mutant. Right panel: Intensity profiles of the dorsal cords (150 pixels) in wild-type, *erg-28(gk697770)*, *ddi-1*

*Figure 5 continued on next page*

Figure 5 continued

(ok1468) and ddi-1(ok1468);erg-28(gk697770) mutant animals. Average pixel intensity of SLO-1 in the dorsal cord was quantified. (C) The number and intensity of SLO-1 puncta in wild-type, erg-28(gk697770), ddi-1(ok1468) and ddi-1(ok1468);erg-28(gk697770) mutant animals were quantified. Error bars represent S.E.M. **p=0.0021 erg-28 vs. ddi-1;erg-28, ****p<0.0001, erg-28 vs. ddi-1;erg-28, one way ANOVA with Tukey's post-hoc analysis. (D) Western analysis shows that total SLO-1::GFP amount is reduced in erg-28(gk697770) mutant and partially restored in ddi-1(ok1468);erg-28(gk697770) double mutant. SLO-1::GFP intensity was normalized to the α-tubulin and the relative intensities compared to the wild type were indicated. N2 does not express GFP and is a negative control for western blotting with anti-GFP antibody. Figure 5—source data 1: This file contains raw measurement data and statistical analysis summary for Figure 5.

The following source data and figure supplement are available for figure 5:

Source data 1. Proteasome inhibition and ddi-1 mutation partially restore SLO-1 expression in erg-28 mutant.
Figure supplement 1. Co-localization of SLO-1 and ERG-28 in the ventral cord.

SLO-1 channel function, we asked whether *erg-28* mutants exhibit an ethanol-resistant behavioral response. *erg-28* mutants exhibit significant ethanol-resistant locomotory behavior, albeit to a lower degree than *slo-1* loss-of-function mutants. This suggests that there is a quantitative correlation between the degree of alcohol sensitivity and the amount of SLO-1 expression at presynaptic terminals. Next, we asked whether increasing SLO-1 at the presynaptic terminals restores the alcohol sensitivity of *erg-28* mutant. While the alcohol sensitivity of the *ddi-1* mutant was not different from the wild type, *ddi-1;erg-28* double mutants were more sensitive than *erg-28* single mutants. Thus, the increased SLO-1 channels in the *ddi-1;erg-28* double mutant are functional and 50% of SLO-1 is sufficient for normal alcohol sensitivity (*Figure 6*, *Videos 2–4*).

## Discussion

The amount and duration of calcium influx into presynaptic terminals has a significant impact on neurotransmitter release by controlling synaptic vesicle docking and fusion. Calcium influx is precisely controlled by several layers of regulatory mechanisms. The BK channel is one of the important negative regulators of calcium influx. The efflux of $K^+$ repolarizes the synaptic terminal, thereby limiting the duration of voltage-gated calcium channel activity. The BK channel current is impacted by many factors, including channel-associated auxiliary factors, post-translational channel modifications, and the number of the channels expressed at the plasma membrane. Our study demonstrates that the density of BK channels at presynaptic terminals influences synaptic transmission and neuronal function, and that ERG-28, an endoplasmic reticulum membrane protein, promotes the transport of SLO-1 BK channels from the ER to the plasma membrane. In *erg-28* mutants, the number of SLO-1 channels at presynaptic terminals is drastically reduced, leading to an alcohol resistance, and the suppression of phenotypic defects of *slo-1(gf)* mutants such as locomotory behavior, synaptic transmission, and altered chemosensory receptor gene expression.

ERG28 was originally identified in yeast as a scaffold in the ER to tether a group of ergosterol synthesizing enzymes, which form a sterol C4 demethylase complex. In higher organisms, however, ERG28 appears to have diverged from yeast ERG28 (*Vinci et al., 2008*). The number of ERG28 transmembrane domains changed from two to four, and its C-terminus acquired a COPI binding consensus sequence, which allows for recycling between the ER and ER-Golgi intermediate compartment (ERGIC). While the involvement of ERG28 in cholesterol synthesis has not been studied in mammals, a recent study in *Arabidopsis* showed that ERG28 tethers C4 demethylase components, but does not interrupt the sterol pathways (*Mialoundama et al., 2013*). Instead, mutation in *ERG28* results in accumulation of cryptic sterol biosynthetic intermediates and consequent inhibition of polar auxin transport. *C. elegans* is not able to synthesize cholesterol de novo, and thus obtains the final sterol products from plants or mammals, such as ergosterol, sitosterol, and cholesterol. In the laboratory, cholesterol is added into the culture medium to maintain *C. elegans*. Substitution of cholesterol with lanosterol, which is a precursor of cholesterol that has a C4 methyl group, does not allow *C. elegans* growth (*Lee et al., 2005*). Furthermore, *C. elegans* does not possess biologically active known cholesterol C4 methyl derivatives (*Mahanti et al., 2014*). Our analysis shows that the

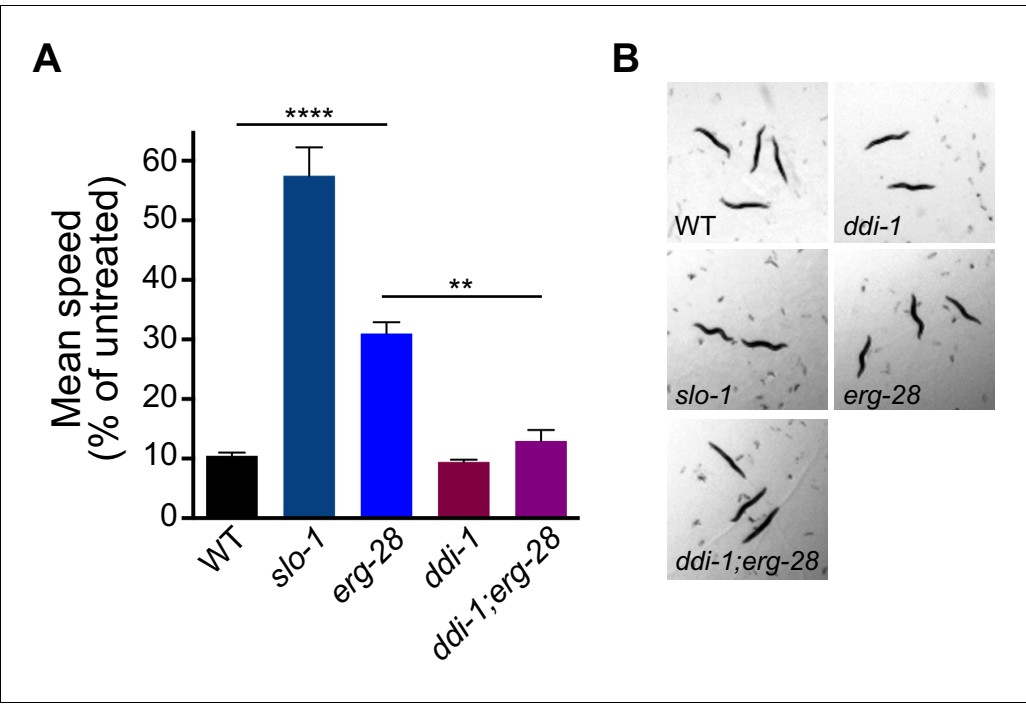

**Figure 6.** *erg-28* mutation confers the resistance to the intoxicating effect of ethanol and *ddi-1* mutation reverses the ethanol-resistant behavior. (**A**) The locomotory speed was measured in the presence of ethanol and its values are divided by the average speed of untreated animals. Error bars represent S.E.M. **p=0.0033, ****p<0.0001, one way ANOVA with Tukey's post-hoc analysis. See *Videos 2–4*. (**B**) The snapshots of animals in the presence of intoxicating animals show sinusoidal posture in *slo-1(eg142lf)*, *erg-28(gk697770)*, *ddi-1(ok1468)*, and *ddi-1(ok1468)*; *erg-28(gk697770)* mutants. *Figure 6—source data 1*: This file contains raw measurement data and statistical analysis summary for *Figure 6*.

The following source data and figure supplement are available for figure 6:

**Source data 1.** *ddi-1* mutation reverses alcohol-resistant locomotion of *erg-28* mutant.

**Figure supplement 1.** A model for ERG-28 function.

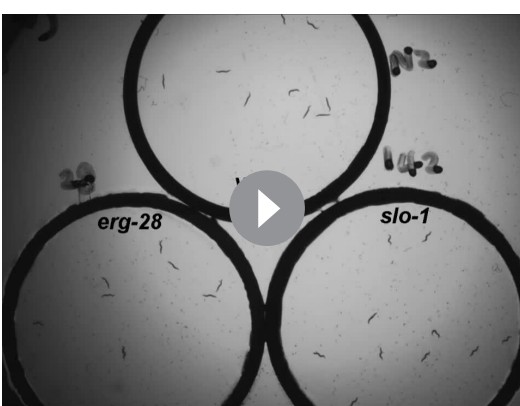

**Video 2.** A movie for wild-type, *erg-28(gk697770)* and *slo-1(eg142)* in the presence of intoxicating dose of ethanol.

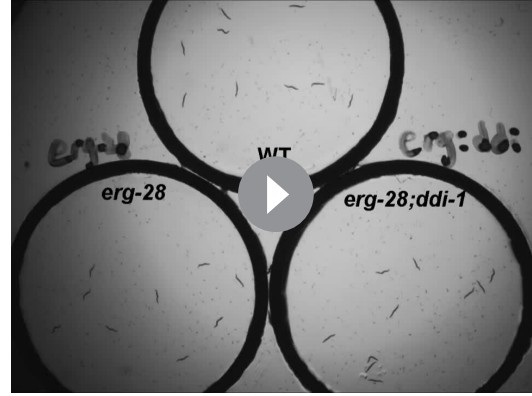

**Video 3.** A movie for wild-type, *erg-28(gk697770)* and *ddi-1(ok1468)*;*erg-28(gk697770)* in the presence of intoxicating dose of ethanol.

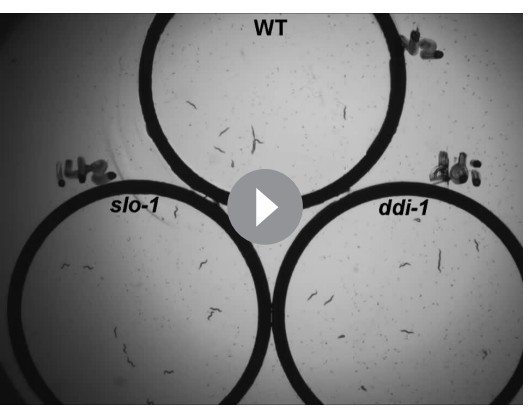

**Video 4.** A movie for wild-type, *slo-1(eg142)* and *ddi-1 (ok1468)* in the presence of intoxicating dose of ethanol. DOI: 10.7554/eLife.24733.027

genes we examined that encode candidate C4 demethylase enzymes do not affect SLO-1 function. While we cannot completely rule out the possibility that distantly related C4 demethylase components redundantly influence SLO-1 function, it is possible that ERG-28 may function independently of sterol synthesis. For instance, it is reported that hERG28 interacts with CLN8, an ER-resident membrane protein involved in neuronal ceroid lipofuscinosis (*Passantino et al., 2013*). CLN8 mutation in humans causes altered levels of sphingolipid and phospholipid. Whether ERG-28 directly mediates SLO-1 channel folding and trafficking or indirectly fosters the lipid environment conducive to SLO-1 channel folding and trafficking needs to be further investigated.

In the absence of ERG-28, SLO-1 expression is drastically reduced at presynaptic terminals and muscle excitation sites and inhibition of proteasome function can partially restore the SLO-1 expression and function. This suggests that in the absence of ERG-28 in the ER, functional SLO-1 channels can still be transported to the plasma membrane. Also, in the absence of ERG-28, SLO-1 does not appear to accumulate in cell bodies. Thus, ERG-28 is not an obligatory factor necessary for the ER exit of SLO-1. Plasma membrane-localized channels and receptors are folded and assembled in the ER and this process is closely monitored by the ER resident protein quality control system to ensure that only correctly folded and assembled channels are trafficked forward to the plasma membrane and to avoid aggregation of misfolded hydrophobic transmembrane domains. It is not clearly understood how cells discriminate folding/assembly intermediates from misfolded proteins in ER. In yeast, amino acid permeases require an ER chaperone to ensure that folding intermediates are not targeted for degradation prematurely, thereby facilitating trafficking to the plasma membrane (*Kota et al., 2007*; *Kuehn et al., 1996*). The SLO-1 channel consists of four pore-forming α subunits and auxiliary subunits, and may require a molecular chaperone that aids channel folding and assembly. Our data support the model in which ERG-28 acts as a chaperone for SLO-1 channel folding/ assembly intermediates so that they are not prematurely recognized as misfolded proteins and targeted for degradation (*Figure 6—figure supplement 1*). It is also possible that ERG-28 may participate in SLO-1 channel trafficking. Designated ER proteins for specific ion channel folding/assembly and trafficking are common, and include RIC-3 that promotes nicotinic acetylcholine receptor surface expression (*Halevi et al., 2002*), CALF-1 that facilitates synaptic localization of UNC-2 voltage-gated calcium channels (*Saheki and Bargmann, 2009*), and NLF-1 that regulates axonal localization of NALC channels (*Xie et al., 2013*). Currently, we cannot demonstrate if there is a direct physical interaction between SLO-1 and ERG-28 in vivo. We predict that the physical interaction would be transient and restricted to the folding/assembly intermediates of SLO-1 channels. It is also likely that ERG-28 has other functions independent of SLO-1 channels. For example, ERG-28 is equally distributed to the dendrites and axons (*Figure 3C*), whereas SLO-1 is relatively specific to axons, suggesting its involvement in the trafficking of dendritic membrane proteins. Further studies are needed to identify additional targets of ERG-28 function.

We attempted to evaluate if ERG-28 function is conserved in mammals (*Figure 1—figure supplement 3*). Three out of four transgenic lines were able to partially rescue *erg-28(cim16)* mutation. This partial rescue could be due to the low primary sequence conservation or sub-optimal expression. Further studies are needed to definitively assess the functional conservation between *C. elegans* and mammalian ERG28 proteins.

We identified *ddi-1* as one of the factors that participate in the degradation of SLO-1 in the absence of ERG-28. *ddi-1* is an orthologue of yeast and mammalian DDI1. Molecular functions of DDI are presently poorly understood. However, it appears to have functions in proteasome-dependent protein degradation and general secretion in yeast. The aspartyl proteinase domain of the yeast DDI1 is important for its function as a negative regulator of general secretion. Recently, the

aspartyl proteinase activity of DDI-1 was shown to be required for the activation of the ER-associated SKN-1 isoform during proteasomal dysfunction in *C. elegans* (*Lehrbach and Ruvkun, 2016*). However, its function under normal conditions is not yet known. Further study is needed to determine if DDI-1 participates in the degradation of SLO-1 directly or indirectly in the absence of ERG-28.

The SLO-1 BK channel is a well-known alcohol target mediating physiological and behavioral alcohol responses (*Davies et al., 2003*; *Davis et al., 2014*; *Dopico et al., 1996*). Our imaging data showed that *erg-28* mutation reduces the synaptic SLO-1 levels and confers ethanol-resistant locomotory behavior. The SLO-1 levels and behavioral ethanol response were reversed when *ddi-1* mutation was introduced to *erg-28* mutants. Previous studies indicated that functional alcohol tolerance has a correlation with BK channel levels at the plasma membrane, although the mechanisms by which its levels are controlled appear to involve several distinct steps (*Palacio et al., 2015*; *Pietrzykowski et al., 2008*). Given that our current study shows that SLO-1 levels are controlled by ERG-28/DDI-1 in the ER, it is tempting to speculate that ERG-28/DDI-1 adjusts overall SLO-1 levels depending on cellular stress or physiological necessity.

# Materials and methods

## Worm strains and maintenance

All *C. elegans* strains were cultured on NGM (nematode growth medium) plates seeded with *E. coli* OP50 at 20°C. The following strains were used for this study: N2 (wild-type), *slo-1(eg142lf)*, *slo-1 (ky399gf)*, *erg-28(cim16)*, *erg-28(gk697770)*, *erg-28(tm6168)*, *unc-36(e251)*, *unc-43(n498n1186)*, *kyIs140l[str-2p::GFP]*, *slo-1(cim105[slo-1::gfp])*, *elks-1(cim107[elks-1::fRFP(FusionRed)])*, *ddi-1(ok1468)*, *cimIs39[erg-28p::mCherry::erg-28, odr-1p::GFP]*, and *zxIs6[unc-17p::ChR2(H134R)::YFP, lin-15(+)]*. The complete list of the strains used is provided in *Supplementary file 1*.

## Molecular biology

To generate various constructs, PCR primers were designed to include restriction enzyme recognition sites. Inserts were generated by PCR with N2 genomic DNA as a template using Q5 high fidelity polymerase (NEB). The PCR products were digested with appropriate restriction enzymes and ligated to backbone plasmids. The list of constructs generated for this study is provided in *Supplementary file 1*.

## Transgenic animals

Transgenic animals were constructed as previously described (*Mello and Fire, 1995*). A DNA mixture adjusted to a final concentration of 100 ng/ml with pBluescript DNA was injected into the gonads of hermaphrodites. Transgenic animals were selected based on the presence of an appropriate marker DNA. When transgenes were used to determine co-localization, transgene DNA concentrations (*myo-3* promoter-based constructs: 2 ng/ml and *unc-129* promoter: 2 ng/ml) were kept low to achieve low level expression. Detailed concentrations of individual DNA constructs for transgene were listed in *Supplementary file 1*.

## CRISPR/Cas9 genome editing

We inserted GFP at the 3' end region of the *slo-1* genomic coding sequence (*Figure 4—figure supplement 1*). This method is based on the observation that if one gene undergoes a double strand break (DBS) and subsequent homologous repair, the second gene that is targeted together with the first in the same injection is more likely to undergo 'co-conversion' (*Arribere et al., 2014*). Specifically, we injected into the gonads of wild-type hermaphrodites a DNA mixture that includes a *slo-1* sgRNA construct and a repair GFP plasmid construct, a *dpy-10* sgRNA construct and a *dpy-10* homologous repair oligonucleotide (99 bp). The *dpy-10* sgRNA will lead to a DSB, and its aberrant non-homologous end-joining repair will lead to the Dpy (dumpy) phenotype in F2 homozygous animals. However, if this DSB is repaired by a homologous repair oligonucleotide, which contains a dominant mutation, the resulting F1 animals will exhibit the dominant Rol (roller) phenotype. We picked Rol animals in the F1 and genotyped them for GFP insertion. For a reason we cannot explain, the efficiency of *slo-1* CRISPR and homologous repair was extremely low, even though we tried 10 different *slo-1* sgRNAs and several different repair constructs. We injected over 350 animals,

screened over 1200 F1 Rol lines for 'co-conversion' by PCR, and recovered only one line in which GFP is correctly inserted in-frame into the 9 bp upstream from the end of the *slo-1* gene. The same 'co-conversion' method was used for the insertion of FusionRed to the last exon of the *elks-1* gene (*Figure 4—figure supplement 1*). We chose FusionRed for tagging because, despite its low brightness, it exhibits a monomeric property and low toxicity (*Shemiakina et al., 2012*).

## Western blotting and quantification

Mixed stage worms were lysed and solubilized in SDS lysis buffer (2% SDS, 100 mM NaCl, 10% glycerol, and 50 mM Tris HCl, pH 6.8) with sonication. Worm extracts were separated on SDS-PAGE and transferred onto PVDF membranes. The separated proteins were probed with anti-GFP (Cell Signaling Technology, Danvers, MA, #2956, RRID:AB_10828931) or anti-tubulin (Developmental Studies Hybridoma Bank, AA4.3, RRID:AB_579793) antibodies. Pixel intensities of GFP and tubulin bands were quantified using Adobe Photoshop CC 2015 (RRID:SCR_014199).

## Microscopy

On day 1 (20–24 hr post L4), adult animals were immobilized on a 2% agarose pad with a 6 mM levamisole solution in M9 buffer (*Oh and Kim, 2013*). Images were acquired using a 63x/1.4 numerical aperture on a Zeiss microscope (Axio-Observer Z1). Images were captured with a CoolSNAP HQ2 CCD camera (Photometrics) or Zyla 4.2 plus (Andor) using the same settings (fluorescence intensity, exposure time, and gain) for a given set of data (35). All of the imaging sessions included wild-type controls to ensure that different results obtained in mutants were not due to changes in illumination. A line-scanning method in Metamorph (RRID:SCR_002368) was used for quantifying the average fluorescence intensity. In each image, a pixel intensity of 150 pixels was measured, followed by subtraction of background intensity from adjacent 150 pixels.

## Aldicarb assay

Aldicarb stock was made at 100 mM in 70% ethanol. NGM agar plates with 1 mM aldicarb were prepared at least 1 day before the assay. 20 hr before the assay, L4 worms were picked. Before the assay three copper rings were placed on the plates and added a drop of OP50 in the center of the rings. When the plates were dry after about 30 min, 20 to 30 worms were placed and examined their paralysis every 10 min. Animals are considered paralyzed when they failed to respond to prodding with platinum wire. Paralyzed worms were removed from the plates and recorded. Assays were repeated three times.

## Electrophysiology

Electrophysiology recordings from the *C. elegans* neuromuscular junction were performed as previously described (*Richmond, 2006*). Evoked currents were recorded in a body wall muscle after eliciting neurotransmitter release from motor neurons carrying channelrhodopsin-2 by a 10 ms illumination of a 470 nm LED (*Liewald et al., 2008*).

## Bortezomib treatment

Bortezomib (LC Laboratories, MA, B-1408) solubilized in DMSO was added to OP50 bacteria seeded NGM agar plates at a final concentration of 20 µM and let the plates absorb the solution. L4 stage worms were transferred to the bortezomib-containing plates and grew them for 24 hr before imaging the worms.

## Ethanol-resistant locomotory behavior assay

Day one adults (24–30 hr post L4) were placed on NGM agar plates with 400 mM ethanol without food. We included wild-type and *slo-1(lf)* mutant animals in the ethanol assays as positive and negative controls. Video frames from three different genotypes were simultaneously acquired from a dissecting microscope fitted with GO-3 camera (QImaging) with 500 ms intervals for 2 min. For each genotype (10 animals each), the experiments were repeated at least three times. We calculated the average speed of the tested animals using the Tracking Objects option in Image-Pro Plus 7.0 (Media Cybernetics). When two animals separate after collision or intersection, new tracks are automatically assigned. This could generate more than 10 tracks for 10 animals.

## Statistical analysis

We used Prism six to perform statistical analysis. Sample number, p value, and statistical test used are indicated in the figure legends. Sample numbers represent independent biological replicates. All of the raw data and statistical analysis summaries are included in the source data file.

## Acknowledgements

Some strains were provided by the CGC, which is funded by NIH Office of Research Infrastructure Programs (P40 OD010440), and by the National BioResource Project in Japan. The NIH, its employees, and officers do not endorse or recommend any commercial products, processes, or services.

## Additional information

### Funding

| Funder | Author |
|---|---|
| National Institute of General Medical Sciences | Chiou-Fen Chuang |
| National Institute on Alcohol Abuse and Alcoholism | Hongkyun Kim |
| Alfred P. Sloan Foundation | Chiou-Fen Chuang |

The funders had no role in study design, data collection and interpretation, or the decision to submit the work for publication.

### Author contributions

KHO, Conceptualization, Data curation, Formal analysis, Investigation, Writing—original draft, Writing—review and editing; JJH, XW, Data curation, Formal analysis, Investigation; C-FC, JER, Data curation, Formal analysis, Investigation, Writing—review and editing; HK, Conceptualization, Data curation, Formal analysis, Supervision, Funding acquisition, Investigation, Writing—original draft, Writing—review and editing

### Author ORCIDs

Hongkyun Kim, http://orcid.org/0000-0002-4879-7122

## Additional files

### Supplementary files

• Supplementary file 1. Lists of plasmid constructs and strains used in this study.

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
