## [Decision Letter]

[Editors’ note: a previous version of this study was rejected after peer review, but the authors submitted for reconsideration. The first decision letter after peer review is shown below.]

Thank you for submitting your work entitled "ERG-28 controls BK channel trafficking in the ER to regulate synaptic function and alcohol response in *C. elegans*" for consideration by *eLife*. Your article has been reviewed by two peer reviewers, and the evaluation has been overseen by a Reviewing Editor and a Senior Editor. The reviewers have opted to remain anonymous.

Our decision has been reached after consultation between the reviewers. Based on these discussions and the individual reviews below, we regret to inform you that your work can not presently be considered for publication in *eLife*.

The reviewers and editors appreciate the potential importance of the findings but felt that a critical mechanistic aspect of ERG-28 function was not properly addressed. Experiments to address this concern are straight-forward, but require more than the standard 2 month revision time that we allow for revisions. We therefore need to reject the manuscript in its present form, but encourage you to submit the work as a new manuscript should you choose to conduct these experiments.

The specific issue that the reviewers identified relate to the question of whether *erg-28* indeed localizes to and presumably acts in the ER (versus the Golgi). A key concern was the observed ER localization may be an artefact due to overexpression. The colocalization with markers were also not convincing. The two key experiments that would alleviate these concerns are:a) rescue the mutant phenotype with a fluorescent protein-tagged version of the protein, (using low copy transgenes to minimize "clogging" up the secretory pathway. b) improve the ER vs. Golgi analysis by expressing the protein in muscle where subcellular structures are easier to localize.

Reviewer #1:

In this work, Oh and colleagues describe the characterization of a novel genetic interactor of SLO-1, ERG-28, the sole *C. elegans* ortholog of Yeast ERG28. Yeast ERG28 acts as a molecular scaffold for sterol biosynthesis enzymes but it's role in animals has not been elucidated yet. The authors propose a model in which ERG-28 controls BK potassium channel trafficking in the ER to regulate synaptic function. While the data presented here is generally consistent with this hypothesis, experimental limitation fail to lend strong enough support to the proposed mechanism.

1) A central hypothesis of this work is that ERG-28 is an ER-resident protein, but the experimental strategy to demonstrate this key aspect is not conclusive. Two different fusions proteins to ERG-28 are used (mCherry and GFP). In both cases, it is not precisely established that these protein fusions are functional. This is a crucial point since tagging such a small protein with a much larger fluorescent molecule can have strongly deleterious effects on protein folding and potentially subcellular localization. In addition, transgenic expression with extrachromosomal arrays almost always leads to over-expression, which for transmembrane proteins can lead to erroneous accumulation of proteins in the ER. Finally, localizing transmembrane proteins to ER versus Golgi compartments is a difficult task in *C. elegans* neurons given their small size. Indeed, diffraction limited microscopy used here cannot resolve compartments so closely juxtaposed within neuronal processes. These experiments could for example be repeated in body wall muscle cells where subcellular compartments can be more clearly distinguished. In the absence of a clear demonstration that ERG-28 is indeed an ER-resident protein, the same data could also support other interpretations, e.g. ERG-28 could be required for the stability of SLO-1 channels at the cell surface by modulating channel recycling instead of promoting trafficking from the ER.

2) Another experimental pitfall is that almost none of the phenotypes and effects described in this manuscript are properly rescued. In Figure 1, only one transgenic line is provided for each condition, when it is standard in the field to analyze multiple independent lines. In particular, absence of rescue in muscles in a single line does not allow the conclusion that the focus of action of ERG-28 is neuronal and not muscular. Along the same lines, in the absence of a proper rescue experiment, it is essential to analyze at least two different alleles to demonstrate that the observed phenotype is indeed due to mutations in *erg-28* and not unknown background mutations. This is particularly a concern for Million Mutation project alleles which are heavily mutagenized.

3) ERG-28 is an interesting new player in the regulation of this major class of potassium channels. However, it's primary sequence conservation with vertebrate ERG-28 orthologs is very low (and not discussed); it will be necessary to specifically verify whether it also regulates vertebrate BK channels before it's functional role can be generalized.

Major comments (detailed):

1) Subcellular localization of ERG-28

I have multiple issues with the data described in Figure 3 to support the subcellular distribution of ERG-28.

Two different fusions proteins to ERG-28 are used (mCherry and GFP). Where and how was the fluorescent protein fused to ERG-28? In both cases, it is not precisely established that these protein fusions are functional. This is a crucial point since tagging such a small protein with a much larger fluorescent molecule can have strongly deleterious effects on protein folding and potentially subcellular localization.

Transgenic expression with extrachromosomal arrays often leads to over-expression, which for transmembrane proteins can lead to erroneous accumulation of proteins in the ER. This is a major concern here when the functional role of ERG-28 is proposed to be in the ER. Ideally, a knockin strategy using small epitope tags should be explored to respect the physiological expression levels of ERG-28. Such a knockin strain should then be functionally validated.

Figure 3 is used to argue that ERG-28 is not present in the Golgi. By barely increasing the brightness of the image, a clear "colocalisation" can be seen in the small "dot" on the right. Increasing brightness also increases colocalisation in the "large blob".

The experiment in Figure 3 is used to strengthen the argument that ERG-28 is indeed a ER resident protein. The strong decrease in fluorescence cannot be linked directly to the removal of this predicted consensus sequence. It might just as well be due to a transgene with low expression levels. In fact, one could expect to see ERG-28-GFP present in compartments beyond the reticulum or even at the plasmamembrane in the absence of a retrieval sequence.

Why were the DA/DB neurons chosen to study ERG-28 localization? No data is provided to demonstrate that *erg-28* is expressed in these neurons.

In general, localizing transmembrane proteins to ER or Golgi compartments is a difficult task in *C. elegans* neurons given their small size. Indeed, diffraction limited microscopy used here can never resolve compartments so closely juxtaposed in neuronal processes. Along these lines, in Figure 5—figure supplement 1, the authors argue that mCherry::ERG-28 and SLO-1::GFP interact transiently because of partial colocalization of the fluorescence signals. Given that these images were acquired with an epifluorescence microscope, it seems very hazardous to conclude to a bona fide colocalisation. In fact, the localization of mCherry::ERG-28 seems qualitatively different in Figure 5—figure supplement 1 and Figure 3, raising further concerns about the functional expression of ERG-28 fusion proteins.

To demonstrate the subcellular localization and associated functional role of ERG-28, an alternative is to perform these experiments in the body wall muscle cells. SLO-1::GFP fluorescence levels could then be used as a functional readout to validate ERG-28 fusion proteins. The signal/noise ratio seems much more favorable in this tissue (Figure 4). The sensitivity of the fluorescence measurement could be validated by comparing homozygous SLO-1::GFP to SLO-1::GFP/+. And the structure of the ER and Golgi compartments are drastically different in these large cells and can be easily distinguished with published fluorescent markers.

2) Almost none of the phenotypes and effects described in this manuscript are properly rescued. In Figure 1, only one line is provided for each condition (muscle vs. neuron) when it is standard in the field to analyze multiple independent transgenic lines. This is particularly an issue here since the absence of rescue with the muscle-expressed line is used as an argument to support the notion that ERG-28 acts pre-synaptically and not post-synaptically. Along the same lines, in the absence of a proper rescue experiment, it would be essential to score at least two different alleles to demonstrate that the observed phenotype is indeed due to mutations in *erg-28* and not unkown background mutation. This is particularly a concern for Million Mutation project alleles which have been heavily mutagenized.

3) Materials and methods: there is little to no information about molecular biology and exact genotypes of strains. Given the absence of length restrictions in this journal, this lack of detail is incomprehensible.

4) *erg-28* is part of an operon with a gene involved in proteasome function (proteasome activator subunit 4). Given the model involving SLO-1 degradation proposed by the authors, this should be discussed. For example, could we expect that the missense mutation in *erg-28(gk697770)* would affect the mRNA stability of the first gene of the operon, causing ERG-28-independent effects on proteasome function?

Reviewer #2:

Oh et al. advance molecular understanding of how proper levels of the BK potassium channel are regulated in neurons. This finding is important because the BK channel has vital roles in physiology, development, behavior, and drug response. They find evidence that the ER protein ERG-28 protects the BK channel against protease degradation. One of these proteases appears to be DDI-1. They visualized subcellular endogenous tagged SLO-1 in context with active zones by the endogenous tagged ELKS-1 protein. CRISPR has been extremely challenging to insert functional fluorophores in *C. elegans*, so they deserve kudos. The study could not have been accomplished without these strains. I only have a few concerns to clarify the role of ERG-28.

1) Does ERG-28 do anything else? For instance, does it modulate abundance of other proteins with consequences on behavior? This may explain why the *erg-28;slo-1(gf)* mutant is hypersensitive to aldicarb, and the *erg-28* single mutant is hypersensitive relative to the *slo-1(lf)* mutant. This could be tested by assaying behavioral, aldicarb, and electrophysiological phenotypes for the *erg-28;slo-1(lf)* double mutant vs. *slo-1* and *erg-28* single mutants.

2) Likewise, it would be good to test whether another membrane protein(s) changes abundance via COTI-dependent interaction via ERG-28.

3) Does *erg-28* overexpression yield excess SLO-1 in the membrane? Overexpression of *slo-1* has consequences for all developmental, behavioral and physiological phenotypes in the manuscript. A positive result would be interesting because it may suggest that basal degradation of SLO-1 is poised to influence these phenotypes.

[Editors’ note: what now follows is the decision letter after the authors submitted for further consideration.]

Thank you for resubmitting your work entitled "ERG-28 controls BK channel trafficking in the ER to regulate synaptic function and alcohol response in *C. elegans*" for further consideration at *eLife*. Your revised article has been favorably evaluated by Richard Aldrich as the Senior Editor, Oliver Hobert as the Reviewing Editor and two reviewers.

The manuscript has been much improved through additional experiments and is almost ready for acceptance now. There are some remaining issues that need to be addressed before acceptance, as outlined below:

1) It is still confusing whether the GFP-tagged ERG-28 transgene shown to restore speed to *slo-1(gf);cim16* in Figure 1 was the *same exact transgene* used to investigate neuron expression in Figure 3 as is suggested in the first paragraph of the subsection “ERG-28 is an ER resident membrane protein expressed in neurons and muscles”. This is a subtle but important issue because if the GFP::ERG-28 transgene was not used in Figure 1, there may be no evidence that it functions in vivo with SLO-1. The two rescued strains appear to use different promoters: H20 panneuronal for Figure 1 and Punc-129 for Figure 3. Please clarify because one transgene might express at different levels or be regulated differently that may affect whether it accumulates in ER.

2) Because it is important that the reader understand that fluorophore-tagged proteins could rescue behavior and expression level of the BK channel, I recommend that all figures, not just legends, note this more obviously rather than solely note the transgene code.

3) Wouldn't it be clearer to state "Loss of *erg-28* suppressed.." rather than "*erg-28* mutation suppresses…" in the Abstract?

4) Figure 6 shows that the *erg-28* mutant displays moderate resistance to intoxication, but nowhere near the strong resistance shown by *slo-1(lf)*. So the statement in the Abstract "*erg-28* mutation…confers ethanol resistant locomotory behavior that resembles *slo-1* loss-of-function mutants." should be corrected.

5) Please do not italicize "and" in the subsection “Worm strains and maintenance”.

---

## [Author Response]

[Editors’ note: the author responses to the first round of peer review follow.]

[…] The specific issue that the reviewers identified relate to the question of whether erg-28 indeed localizes to and presumably acts in the ER (versus the Golgi). A key concern was the observed ER localization may be an artefact due to overexpression. The colocalization with markers were also not convincing. The two key experiments that would alleviate these concerns are:a) rescue the mutant phenotype with a fluorescent protein-tagged version of the protein, (using low copy transgenes to minimize "clogging" up the secretory pathway.

In the original manuscript previously submitted, the rescue experiment (locomotion speed measurement in Figure 1) was performed with GFP-tagged ERG-28, but we mistakenly omitted this information. We now provide detailed genotypes in the Figures. We also provide Supplementary Table (an Excel file) with a complete list of strains, their genotypes, and plasmid constructs used in this study.

To further demonstrate that fluorescent protein-tagged ERG-28 is functional, we show that expression of mCherry::ERG-28 driven by its own promoter restores SLO-1::GFP expression in neurons and muscles (Figure 4).

Thus, we demonstrate that both GFP::ERG-28 and mCherry::ERG-28 are functional.

*b) improve the ER vs. Golgi analysis by expressing the protein in muscle where subcellular structures are easier to localize.*

This was an excellent suggestion. We have now examined ERG-28 subcellular localization in muscle cells. Given that GFP-tagged ERG-28 is functional, we expressed GFP::ERG-28 together with either the ER marker PISY-1::mCherry or the Golgi maker AMAN-2::mCherry in muscle cells. To ensure a low level of expression, we used a low concentration of the expression construct *myo- 3p::gfp::erg-28* and *myo-3p::PISY-1::mCherry* (or *AMAN-2::mCherry*). As the reviewer pointed out, ER and Golgi exhibit completely different patterns in muscle cells. GFP::ERG-28 does not co-localize with the Golgi marker. By contrast, GFP::ERG-28 well co-localizes with PISY-1::mCherry (Figure 3).

We imaged body wall muscle cells by acquiring Z-stack images from the muscle interior (the muscle belly) to the muscle plasma membrane. If native localization of GFP::ERG-28 is the plasma membrane rather than ER, one would expect that GFP::ERG-28 signals would be strong in the stacks that represent the plasma membrane. However, our cross comparison with the ER marker shows that GFP::ERG-28 signals were not high in the stacks representing the plasma membrane, indicating that ERG-28 localizes in the ER (Figure 3—figure supplement 1). In addition to these new results, ERG-28 has a well-defined classical ER retention/retrieval sequence motif, -KKXX-COOH, which is also called the dilysine motif. In fact, other studies have shown that this dilysine motif in the C-terminus is critical for determining ER localization. For example, a Golgi protein can be redistributed to the ER by attaching this dilysine motif to its C- terminus (Gorleku et al.). Another example is that a gap junction protein can be trapped in the ER by attaching this motif (Maza et al.). The mutant, *cim16*, that we isolated in our unbiased genetic screen has a mutation that deletes 7 C- terminal amino acid residues. The phenotype of this mutant is comparable to that of another allele *gk697770* that has a nonsense mutation in the beginning of the coding sequence.

Gorleku, O.A., Barns, A.-M., Prescott, G. R. Greaves, J., and Chamberlain, L. H. (2011). Endoplasmic Reticulum Localization of DHHC Palmitoyltransferases Mediated by Lysine-based Sorting Signals. J. Biol. Chem. 286:39573-39584.

Maza, J., Das Sarma, J., and Koval, M.(2005). Defining a minimal motif to prevent connexin oligomerization in the endoplasmic reticulum. J. Biol. Chem. 280:21115-21121.

Reviewer #1:

In this work, Oh and colleagues describe the characterization of a novel genetic interactor of SLO-1, ERG-28, the sole C. elegans ortholog of Yeast ERG28. Yeast ERG28 acts as a molecular scaffold for sterol biosynthesis enzymes but it's role in animals has not been elucidated yet. The authors propose a model in which ERG-28 controls BK potassium channel trafficking in the ER to regulate synaptic function. While the data presented here is generally consistent with this hypothesis, experimental limitation fail to lend strong enough support to the proposed mechanism.

1) A central hypothesis of this work is that ERG-28 is an ER-resident protein, but the experimental strategy to demonstrate this key aspect is not conclusive. Two different fusions proteins to ERG-28 are used (mCherry and GFP). In both cases, it is not precisely established that these protein fusions are functional. This is a crucial point since tagging such a small protein with a much larger fluorescent molecule can have strongly deleterious effects on protein folding and potentially subcellular localization. In addition, transgenic expression with extrachromosomal arrays almost always leads to over-expression, which for transmembrane proteins can lead to erroneous accumulation of proteins in the ER. Finally, localizing transmembrane proteins to ER versus Golgi compartments is a difficult task in C. elegans neurons given their small size. Indeed, diffraction limited microscopy used here cannot resolve compartments so closely juxtaposed within neuronal processes. These experiments could for example be repeated in body wall muscle cells where subcellular compartments can be more clearly distinguished. In the absence of a clear demonstration that ERG-28 is indeed an ER-resident protein, the same data could also support other interpretations, e.g. ERG-28 could be required for the stability of SLO-1 channels at the cell surface by modulating channel recycling instead of promoting trafficking from the ER.

Fluorescent protein-tagged ERG-28 transgenes are functional. We clarified the text to reflect the fact that the rescued locomotory phenotype in Figure 1 was achieved using GFP-tagged ERG-28. In Figure 4, we now also show that expression of mCherry::ERG-28 restores the SLO-1::GFP levels in neurons and muscles.

As the reviewer recommended, we determined ERG-28 localization in muscle cells and the results (Figure 3 and Figure 3—figure supplement 1) are consistent with ER localization. This was also discussed in the section above.

Over-expression of plasma membrane proteins can lead to erroneous localization in ER. However, it does not lead to complete ER mislocalization. For example, overexpression of SLO-1 by transgenic arrays does not lead to complete mislocalization of SLO-1 to the ER; SLO-1 does localize at the plasma membrane but it can be found in the ER in addition. The co- localization of GFP::ERG-28 with mCherry::PISY-1 in different z-sections at the same intensity ratio (Figure 3—figure supplement 1) and the presence of the well-characterized ER retention/retrieval motif strongly support the idea that ERG-28 functions in the ER.

2) Another experimental pitfall is that almost none of the phenotypes and effects described in this manuscript are properly rescued. In Figure 1, only one transgenic line is provided for each condition, when it is standard in the field to analyze multiple independent lines. In particular, absence of rescue in muscles in a single line does not allow the conclusion that the focus of action of ERG-28 is neuronal and not muscular. Along the same lines, in the absence of a proper rescue experiment, it is essential to analyze at least two different alleles to demonstrate that the observed phenotype is indeed due to mutations in erg-28 and not unknown background mutations. This is particularly a concern for Million Mutation project alleles which are heavily mutagenized.

In our original manuscript, we showed the suppression of the *slo-1(ky399gf*) phenotype by two *erg-28* alleles, *cim16* and *gk697770. cim16* is an allele isolated in this study and *gk697770* is from the Million Mutation project. In our revised manuscript, we now show that one additional allele, *tm6168*, also suppresses the locomotory phenotype of *slo-1(ky399gf)* mutant animals (Figure 1 and Figure 1—figure supplement 2). The *tm6168* allele is a knock- out deletion/insertion allele generated by the BioResource Project in Japan.

In Figure 1, we added the additional transgenic rescue lines for muscle and neurons. These additional transgenic lines showed the same results as our previous results.

3) ERG-28 is an interesting new player in the regulation of this major class of potassium channels. However, it's primary sequence conservation with vertebrate ERG-28 orthologs is very low (and not discussed); it will be necessary to specifically verify whether it also regulates vertebrate BK channels before it's functional role can be generalized.

A previous paper by another group identified *C. elegans erg-28* as a yeast ERG28 homolog and this was referenced in the manuscript. Furthermore, we also now include information on homology of *erg-28* with the human homolog C14orf1 in the text as well as Figure 1—figure supplement 1.

Since studies of vertebrate BK channel trafficking with C14orf1 require completely different sets of experimental system and analysis, it is beyond the scope of this manuscript. As requested by the reviewer, we determined whether C14orf1 could replace ERG-28 function in *C. elegans*. Specifically, we performed a rescue experiment of *erg-28(cim16) slo-1(ky399gf)* double mutant animals with the human homolog C14orf1. We generated four different transgenic lines that express GFP-tagged C14orf1 (human ERG28) under the control of the pan-neuronal *H20* promoter and found that three lines exhibit partial rescue (reverting the speed of *erg-28(cim16) slo-1(ky399gf)* double mutant to that of the *slo-1(ky399gf)* mutant), suggesting functional conservation. These results are now included in Figure 1—figure supplement 3.

Major comments (detailed):

1) Subcellular localization of ERG-28

I have multiple issues with the data described in Figure 3 to support the subcellular distribution of ERG-28.

Two different fusions proteins to ERG-28 are used (mCherry and GFP). Where and how was the fluorescent protein fused to ERG-28? In both cases, it is not precisely established that these protein fusions are functional. This is a crucial point since tagging such a small protein with a much larger fluorescent molecule can have strongly deleterious effects on protein folding and potentially subcellular localization.

We have addressed below.

Transgenic expression with extrachromosomal arrays often leads to over-expression, which for transmembrane proteins can lead to erroneous accumulation of proteins in the ER. This is a major concern here when the functional role of ERG-28 is proposed to be in the ER. Ideally, a knockin strategy using small epitope tags should be explored to respect the physiological expression levels of ERG-28. Such a knockin strain should then be functionally validated.

Because fluorescent protein-tagged ERG-28 can rescue the locomotory phenotype of *erg-28 slo-1(gf)* double mutant animals (Figure 1) and SLO-1 levels in *erg-28* mutant (Figure 4), we proceeded to the localization study with these transgenes.

Figure 3 is used to argue that ERG-28 is not present in the Golgi. By barely increasing the brightness of the image, a clear "colocalisation" can be seen in the small "dot" on the right. Increasing brightness also increases colocalisation in the "large blob".

To address this point, we examined the colocalization of ERG-28 and the Golgi marker in muscle cells, where Golgi and ER expression patterns are easier to differentiate. We found that ERG-28 does not co-localize with the Golgi marker. These new data are presented in Figure 3 and Figure 3—figure supplement 1.

The experiment in Figure 3 is used to strengthen the argument that ERG-28 is indeed a ER resident protein. The strong decrease in fluorescence cannot be linked directly to the removal of this predicted consensus sequence. It might just as well be due to a transgene with low expression levels. In fact, one could expect to see ERG-28-GFP present in compartments beyond the reticulum or even at the plasmamembrane in the absence of a retrieval sequence.

Figure 3 demonstrates that the C-terminal 7 amino acid deletion, which occurs in the *cim16* allele that we isolated in this study, results in defective expression and/or localization, and therefore loss of function. We cannot determine if the functional defect of the *cim16* allele is due to a structural instability independently of defective ER localization or a localization defect caused by the absence of the ER retrieval sequence. We have discussed this in the text.

Why were the DA/DB neurons chosen to study ERG-28 localization? No data is provided to demonstrate that erg-28 is expressed in these neurons.

We chose the DA/DB neurons to determine ERG-28 localization because their cell bodies and dendrites are located in the ventral cord, whereas axon terminals are located in the dorsal cord. This clear separation allows us to determine the involvement of ERG-28 in polarity (expression in dendrites or axons) as well as subcellular localization. These are mentioned in the text.

We did not directly examine whether *erg-28* is expressed in the DA/DB neurons. The dorsal cord in the body consists of axons from only five motor neurons (DA, DB, DD, VD, and AS). If *erg-28* is not expressed in DA and DB neurons, our analysis of the dorsal cord in *erg-28* mutants should show two fifths (2 out of 5 neurons) of the SLO-1::GFP puncta as bright as those of wild-type animals (Figure 5). That was not the case, indicating that DA/DB neurons also express *erg-28*.

In general, localizing transmembrane proteins to ER or Golgi compartments is a difficult task in C. elegans neurons given their small size. Indeed, diffraction limited microscopy used here can never resolve compartments so closely juxtaposed in neuronal processes. Along these lines, in Figure 5—figure supplement 1, the authors argue that mCherry::ERG-28 and SLO-1::GFP interact transiently because of partial colocalization of the fluorescence signals. Given that these images were acquired with an epifluorescence microscope, it seems very hazardous to conclude to a bona fide colocalisation. In fact, the localization of mCherry::ERG-28 seems qualitatively different in Figure 5—figure supplement 1 and Figure 3, raising further concerns about the functional expression of ERG-28 fusion proteins.

We have now examined the localization of ERG-28 in body wall muscle cells. We agree with the reviewer that we cannot conclude that the occasional observation of the overlap of the GFP and mCherry signal is a bona fide co- localization. In the text, we state that “we find that SLO-1::GFP puncta rarely co-localize with mCherry::ERG-28 although potential co-localization can be occasionally observed in neuronal cell bodies (Figure 5—figure supplement 1).” In Figure 3, GFP::ERG-28 was expressed in a subset of motor neurons, whereas in Figure 5—figure supplement 1, mCherry::ERG-28 was expressed using its own promoter, which appears to be active in most, if not all, of the neurons.

To demonstrate the subcellular localization and associated functional role of ERG-28, an alternative is to perform these experiments in the body wall muscle cells. SLO-1::GFP fluorescence levels could then be used as a functional readout to validate ERG-28 fusion proteins. The signal/noise ratio seems much more favorable in this tissue (Figure 4). The sensitivity of the fluorescence measurement could be validated by comparing homozygous SLO-1::GFP to SLO-1::GFP/+. And the structure of the ER and Golgi compartments are drastically different in these large cells and can be easily distinguished with published fluorescent markers.

The points raised in these paragraphs are addressed above.

2) Almost none of the phenotypes and effects described in this manuscript are properly rescued. In Figure 1, only one line is provided for each condition (muscle vs. neuron) when it is standard in the field to analyze multiple independent transgenic lines. This is particularly an issue here since the absence of rescue with the muscle-expressed line is used as an argument to support the notion that ERG-28 acts pre-synaptically and not post-synaptically. Along the same lines, in the absence of a proper rescue experiment, it would be essential to score at least two different alleles to demonstrate that the observed phenotype is indeed due to mutations in erg-28 and not unkown background mutation. This is particularly a concern for Million Mutation project alleles which have been heavily mutagenized.

The points raised in these paragraphs are addressed above.

3) Materials and methods: there is little to no information about molecular biology and exact genotypes of strains. Given the absence of length restrictions in this journal, this lack of detail is incomprehensible.

We have added a list of strains with detailed genotypes and a list of plasmid constructs used in this study to [Supplementary-material SD9-data].

4) erg-28 is part of an operon with a gene involved in proteasome function (proteasome activator subunit 4). Given the model involving SLO-1 degradation proposed by the authors, this should be discussed. For example, could we expect that the missense mutation in erg-28(gk697770) would affect the mRNA stability of the first gene of the operon, causing ERG-28-independent effects on proteasome function?

Transcription from an operon generates a polycistronic pre-mRNA that is co- transcriptionally processed by cleavage and polyadenylation at the 3’ end of each gene, thus generating separate mature mRNAs. It is very unlikely that a point mutation in the downstream gene would have effects on the pre-mRNA processing and stability of the processed mature mRNAs of upstream genes. We added a reference that addresses this. Importantly, if the *erg-28* mutation caused reduction of function in the proteasome activator subunit 4 gene and consequent reduction of proteasome dependent degradation, it may increase, but not decrease, SLO-1 channel levels.

Reviewer #2:

[…] 1) Does ERG-28 do anything else? For instance, does it modulate abundance of other proteins with consequences on behavior? This may explain why the erg-28;slo-1(gf) mutant is hypersensitive to aldicarb, and the erg-28 single mutant is hypersensitive relative to the slo-1(lf) mutant. This could be tested by assaying behavioral, aldicarb, and electrophysiological phenotypes for the erg-28;slo-1(lf) double mutant vs. slo-1 and erg-28 single mutants.

2) Likewise, it would be good to test whether another membrane protein(s) changes abundance via COTI-dependent interaction via ERG-28.

As stated above, we presently do not know of any other membrane proteins whose trafficking is influenced by ERG-28. While this is an interesting possibility, it is beyond the scope of the present study which focuses on the regulation of SLO-1.

3) Does erg-28 overexpression yield excess SLO-1 in the membrane? Overexpression of slo-1 has consequences for all developmental, behavioral and physiological phenotypes in the manuscript. A positive result would be interesting because it may suggest that basal degradation of SLO-1 is poised to influence these phenotypes.

This is an interesting point. We examined a transgenic animal line overexpressing *erg-28,* but we did not observe any measurable increase in SLO-1 channels or uncoordinated locomotory phenotype as seen in the *slo- 1(gf)* mutant. However, due to the limited quantitative resolution of SLO- 1::GFP, it may not be possible to detect a small increase that does not manifest as severely as in *slo-1(gf)* mutant.

[Editors' note: the author responses to the re-review follow.]

The manuscript has been much improved through additional experiments and is almost ready for acceptance now. There are some remaining issues that need to be addressed before acceptance, as outlined below:

1) It is still confusing whether the GFP-tagged ERG-28 transgene shown to restore speed to slo-1(gf);cim16 in Figure 1 was the same exact transgene used to investigate neuron expression in Figure 3 as is suggested in the first paragraph of the subsection “ERG-28 is an ER resident membrane protein expressed in neurons and muscles”. This is a subtle but important issue because if the GFP::ERG-28 transgene was not used in Figure 1, there may be no evidence that it functions in vivo with SLO-1. The two rescued strains appear to use different promoters: H20 panneuronal for Figure 1 and Punc-129 for Figure 3. Please clarify because one transgene might express at different levels or be regulated differently that may affect whether it accumulates in ER.

The pan-neuronal H20 promoter in Figure 1 (rescue of *erg-28;slo-1(gf)* locomotory phenotype) and the *unc-129* promoter in Figure 3 (localization of ERG-28 in neurons) were used to express GFP-tagged ERG-28. We used the *unc-129* promoter in Figure 3 instead of the H20 promoter because we felt that the pan-neuronal H20 promoter was not suitable for investigating subcellular localization of GFP::ERG-28. Pan-neuronal expression of GFP::ERG-28 from the H20 promoter makes it difficult to discern axons, dendrites, and individual cell bodies because of the high density of neurons in the ventral cord. By contrast, because the *unc-129* promoter drives expression only in the DA/DB cholinergic motor neurons, the exact localization of GFP::ERG-28 in axons, dendrites, and cell bodies is easily discernible. The DA/DB neurons innervate the dorsal body wall muscle that is essential for forward and backward movements (As described in our previous response, *erg-28* is likely to be expressed in these neurons). For this reason, the *unc-129* promoter has been widely used to investigate the localization or dynamics of GFP-tagged synaptic proteins by ourselves and others. For example, we previously used the *unc-129* promoter to study presynaptic localization of SLO-1 and the UNC-2 voltage gated calcium channel (Oh et al. BMC Neuroscience (2015) 16:26). Over-expression of the plasma membrane proteins may lead to a minor ectopic ER localization in addition to their normal localization. In the case of ERG-28, we did not find any other localization besides ER with our imaging setup. While we used the H20 and *unc-129* promoters to express GFP-tagged ERG-28 for specific purposes, it is unlikely that these different promoters would make a profound difference to the subcellular localization of ERG-28.

We made a change in the subsection “ERG-28 is an ER resident membrane protein expressed in neurons and muscles”: “Localization of GFP-tagged ERG-28 most likely reflects endogenous ERG-28 localization, since GFP-tagged ERG-28 was able to rescue the loss of ERG-28 (Figure 1).”

2) Because it is important that the reader understand that fluorophore-tagged proteins could rescue behavior and expression level of the BK channel, I recommend that all figures, not just legends, note this more obviously rather than solely note the transgene code.

We made changes in the Figure 1 and Figure 3, so that fluorescent protein-tagged proteins are easily noted.

3) Wouldn't it be clearer to state "Loss of erg-28 suppressed.." rather than "erg-28 mutation suppresses…" in the Abstract?

We agree with the reviewer and changed to “Loss of erg-28 suppressed…” in the Abstract.

4) Figure 6 shows that the erg-28 mutant displays moderate resistance to intoxication, but nowhere near the strong resistance shown by slo-1(lf). So the statement in the Abstract "erg-28 mutation…confers ethanol resistant locomotory behavior that resembles slo-1 loss-of-function mutants." should be corrected.

We have changed the Abstract. Now it reads “conferred significant ethanol-resistant locomotory behavior, resembling *slo-1* loss-of-function mutants, albeit to a lesser extent.”

Because of this change, we had to make some minor adjustment in the Abstract to meet the 150-word limit.

*5) Please do not italicize "and" in the subsection “Worm strains and maintenance”.*

It is corrected.